# Gut microbiota modulates lung fibrosis severity following acute lung injury in mice

Ozioma S. Chioma[1], Elizabeth K. Mallott [2,3], Austin Chapman[1], Joseph C. Van Amburg [1], Hongmei Wu[1], Binal Shah-Gandhi[1], Nandita Dey[1], Marina E. Kirkland[1], M. Blanca Piazuelo [4], Joyce Johnson[5], Gordon R. Bernard [6], Sobha R. Bodduluri [7], Steven Davison[8], Bodduluri Haribabu [7], Seth R. Bordenstein [2,3,5] & Wonder P. Drake [1,5✉]

Independent studies demonstrate the significance of gut microbiota on the pathogenesis of chronic lung diseases; yet little is known regarding the role of the gut microbiota in lung fibrosis progression. Here we show, using the bleomycin murine model to quantify lung fibrosis in C57BL/6 J mice housed in germ-free, animal biosafety level 1 (ABSL-1), or animal biosafety level 2 (ABSL-2) environments, that germ-free mice are protected from lung fibrosis, while ABSL-1 and ABSL-2 mice develop mild and severe lung fibrosis, respectively. Metagenomic analysis reveals no notable distinctions between ABSL-1 and ABSL-2 lung microbiota, whereas greater microbial diversity, with increased *Bifidobacterium* and *Lactobacilli*, is present in ABSL-1 compared to ABSL-2 gut microbiota. Flow cytometric analysis reveals enhanced IL-6/STAT3/IL-17A signaling in pulmonary CD4 + T cells of ABSL-2 mice. Fecal transplantation of ABSL-2 stool into germ-free mice recapitulated more severe fibrosis than transplantation of ABSL-1 stool. *Lactobacilli* supernatant reduces collagen 1A production in IL-17A- and TGFβ1-stimulated human lung fibroblasts. These findings support a functional role of the gut microbiota in augmenting lung fibrosis severity.

[1] Division of Infectious Disease, Department of Medicine, Vanderbilt University School of Medicine, Nashville, TN, USA. [2] Department of Biological Sciences, Vanderbilt University, Nashville, TN, USA. [3] Vanderbilt Microbiome Initiative, Vanderbilt University, Nashville, TN, USA. [4] Division of Gastroenterology, Hepatology and Nutrition Research, Department of Medicine, Vanderbilt University School of Medicine, Nashville, TN, USA. [5] Department of Pathology, Microbiology, and Immunology, Vanderbilt University School of Medicine, Nashville, TN, USA. [6] Division of Pulmonary and Critical Care, Department of Medicine, Vanderbilt University School of Medicine, Nashville, TN, USA. [7] Department of Microbiology and Immunology, University of Louisville School of Medicine, Louisville, KY, USA. [8] Comparative Medicine Research Unit, University of Louisville School of Medicine, Louisville, KY, USA. ✉email: wonder.drake@vumc.org

Accumulating evidence suggests that the gut microbiota have a profound impact on lung pathophysiology, such as the impact of early gut microbiome on childhood asthma[1]. Fibrotic lung disease, a major cause of morbidity and mortality worldwide, occurs when fibroblasts are stimulated to overproduce collagen and other extracellular matrix components in response to inflammatory signals[2]. Current treatments rely on slowing disease progression, yet the mortality rate for some fibrotic lung diseases remains unacceptably high[3–5]. Studies in patient cohorts with idiopathic pulmonary fibrosis (IPF) have described altered lung microbiota and correlate high lung bacterial burden with enhanced disease[6,7]. Additionally, metagenomic sequencing of fecal samples from autoimmune fibrotic lung diseases, such as systemic sclerosis and silicosis, revealed dysbiosis and lack of gut community diversity[8,9]. Despite these striking clinical correlations, mechanistic studies using murine models to define the role of the gut microbiota in pulmonary fibrosis development and progression are missing.

A growing body of literature supports a gut-lung axis, where processes occurring in the gut associate with outcomes of lung pathologies[10,11]. The gut microbiota can influence the lung inflammatory environment in several ways. Production of inflammatory cytokines such as IL-6, which play a role in fibrosis development, can be induced by gut microbes[12–14]. Enhanced IL-6 signaling has been shown to contribute to interstitial lung disease (ILD) pathogenesis, through the IL-6/STAT3/IL-17A signaling pathway[15–17]. Conversely, microbial metabolites, such as the short chain fatty acid (SCFA) butyrate, produced by *Bacteroidetes*, *Clostridia*, and *Lactobacilli* can have anti-inflammatory properties and bias T cells toward a Th1 or T regulatory rather than Th17 phenotype[18,19]. These contrasting roles for the gut microbiota in immune activation suggest that the specific microbial composition of gut communities is important for enhancing or suppressing pathogenic immune responses contributing to lung fibrosis.

This study used direct modulation of the gut microbiota via rearing environments and fecal microbiota transplantation (FMT) in a bleomycin-induced murine model to demonstrate that a functional gut-lung axis modulates lung fibrosis. While germ-free (GF) mice developed distinctively less lung fibrosis than ABSL-1 or ABSL-2 housed mice, FMT of the ABSL-2 stool into germ-free mice before bleomycin administration recapitulated the severe fibrosis phenotype. These results specify that degrees of lung fibrosis severity can be modulated by the absence and/or diversity of the gut microbiota.

## Results

### Rearing environment impacts severity of pulmonary fibrosis.
To assess the contribution of the gut microbiota to lung fibrosis severity, we housed mice in three distinct environments: GF (no microorganisms), ABSL-1 (no experiments involving infectious agents; mice possess only commensal organisms), and ABSL-2 (experiments involving infectious agents of moderate potential hazard to personnel are present in environment). None of the mice in our investigations were ever infected; instead, 6-week-old male and female mice in each environment were administered either saline or bleomycin intranasally and monitored for 14 days. When given intranasal bleomycin, all GF mice exhibited 100% survival (Fig. 1a). ABSL-2 mice that were administered bleomycin exhibited the greatest mortality, and ABSL-1 survival was intermediate of GF and ABSL-2 mice (Fig. 1a). Remarkable distinctions in survival were noted among the ABSL-2 and GF mice ($p = 0.026$, log-rank Mantel-Cox test). When treated with saline intranasally, mice in all housing cohorts had similar survival plots (Fig. S1a), did not experience distinct weight loss (Fig. S1b), and had similar lung collagen content (Fig. S1c), suggesting no acute lung injury. There were notable morbidity distinctions, in terms

of weight loss, to mice administered bleomycin. ABSL-1 and ABSL-2 mice administered saline demonstrated less weight loss than those receiving bleomycin ($p < 0.0001$ and $p < 0.01$, respectively; two-way ANOVA with Tukey's multiple comparison's test), while mice housed in gnotobiotic facilities did not lose weight upon bleomycin treatment (Fig. 1b). Also, GF mice receiving bleomycin exhibited less weight loss compared to ABSL-1 mice ($p < 0.001$, two-way ANOVA with Tukey's multiple comparison's test), as well as ABSL-2 mice ($p < 0.01$, two-way ANOVA with Tukey's multiple comparison's test) (Fig. 1b). Qualitative and quantitative histologic assessments of the lung following bleomycin administration in the three different rearing environments were conducted. Hematoxylin and eosin (H&E) staining, and trichrome staining of right middle lobe lung sections from bleomycin-treated mice under the three rearing conditions were used to determine damage to the lung (Representative images in Fig. 1c). Fibrosis severity was less severe by Ashcroft scoring in GF mice compared to ABSL-1 mice, as well as compared to ABSL-2 mice (Fig. 1d). Notably, quantification of collagen content by the Sircol assay revealed that ABSL-2 mice had notably higher lung collagen content than GF ($p < 0.0001$, two-way ANOVA with Tukey's multiple comparison's test) or ABSL-1 mice ($p < 0.0001$, two-way ANOVA with Tukey's multiple comparison's test). The ABSL-1 mice had intermediate lung fibrosis distinct from GF and ABSL-2 mice ($p < 0.01$, two-way ANOVA with Tukey's multiple comparison's test) (Fig. 1e). Remarkably, the GF mice administered bleomycin had pulmonary collagen contents like the mice administered intranasal saline (Fig. 1e). We confirmed these observations of higher collagen quantities in ABSL-1 and ABSL-2 mice by performing immunoblot analysis for alpha smooth muscle actin (α-SMA) analysis on lung samples acquired 14 days after bleomycin administration. We noted higher α-SMA content in ABSL-2 mice, compared to GF or ABLS-1 mice (Fig. S1d, e). Quantification of immunoblot analysis of α-SMA is depicted in Fig. S1e. These findings indicate that the environment in which the murine host lived contributed to lung fibrosis severity.

### Lack of gut microbial diversity associates with severity of lung fibrosis.
To investigate the hypothesis that the gut microbiota is an important contributor to the differences in fibrosis severity between ABSL-1 and ABSL-2 housed mice, we performed metagenomic analysis on fecal pellets from mice in each housing cohort. We did not detect microorganisms in the stool of GF mice by sequencing and culture, as expected. Shannon alpha diversity, a measure of species richness and evenness, was considerable higher in ABSL-1 mice compared with ABSL-2 mice using a Wilcoxon rank sum test ($W = 166$, $p = 0.01$) (Fig. 2a). Shannon alpha diversity did not differ between treatments (bleomycin vs. saline) using a Wilcoxon rank sum test ($W = 367$, $p = 0.278$), suggesting that bleomycin treatment did not alter the diversity of the gut microbiota. Alpha diversity was also assessed using Pielou's evenness, a measure of species evenness only, and ABSL-1-reared mice were found to have higher Pielou's evenness scores than ABSL-2 mice ($p = 0.009$, Wilcoxon) (Fig. S2a). Species richness does not differ between housing environment (Fig. S2b).

We then compared the relative abundance of phyla using a generalized linear model (GLM) with a negative binomial distribution. ABSL-1 mice had appreciably higher relative abundances of *Proteobacteria* (GLM, $\chi^2 = 20.221$, $p < 0.001$) and *Verrucomicrobia* (GLM, $\chi^2 = 4.487$, $p = 0.034$) (Fig. 2b). Increased proteobacteria abundance has been linked with both inflammatory gut and airway environments, as in Crohn's disease[20], asthma[21] and after lung transplantation[22]. Beta diversity differences between ABSL-1 and ABSL-2 microbiota compositions also differed significantly when

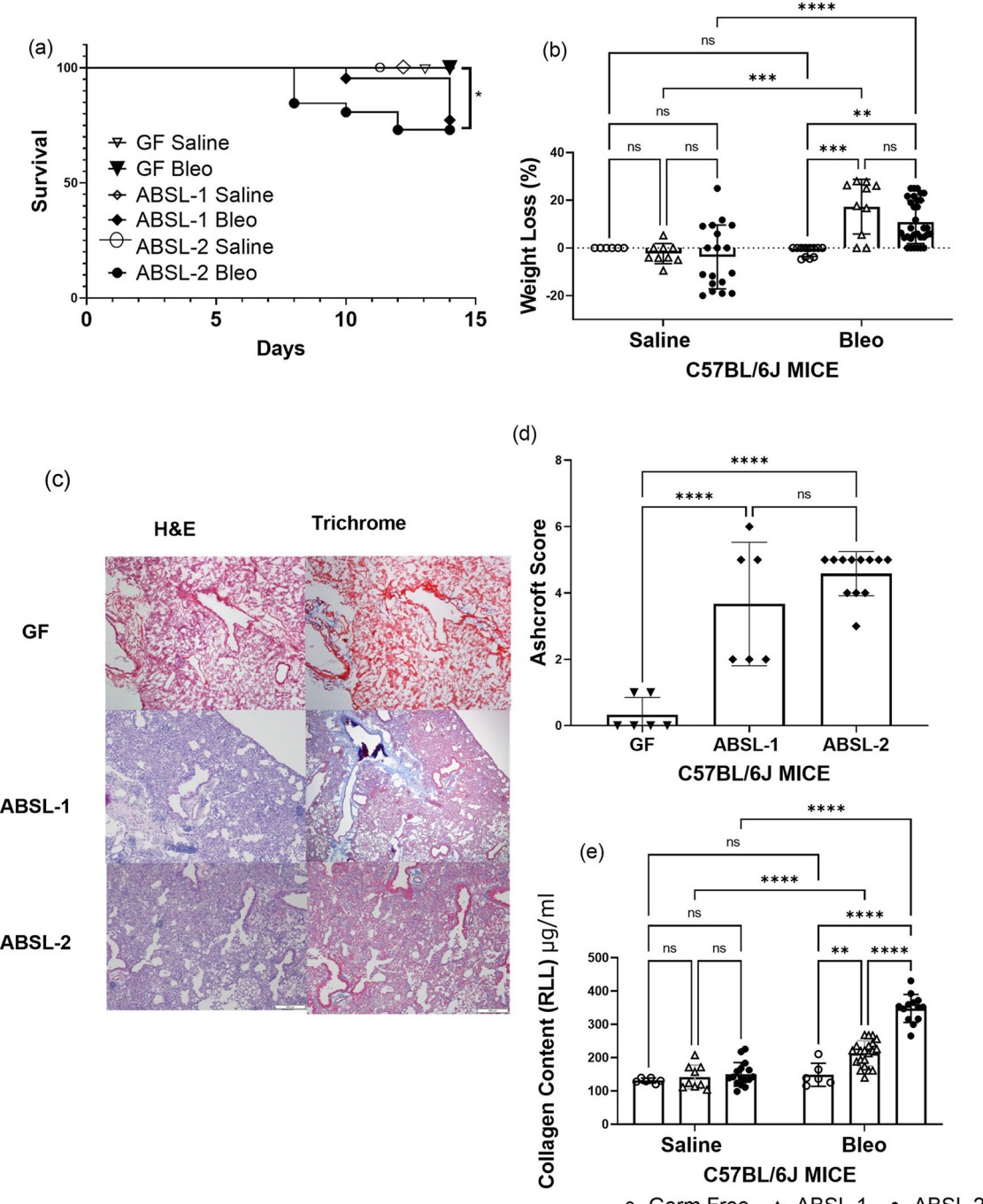

**Fig. 1 Germ-free mice are protected from lung fibrosis morbidity and mortality. a** Survival curve of mice housed in gnotobiotic, ABSL-1, or ABSL-2 facilities and intranasally inoculated with 75 µl containing 0.04 Units of bleomycin. Control animals were intranasally inoculated with 75 µl of saline. **b** Body weight of mice in the housing and treatment cohorts was continuously measured and recorded daily from day 1 to day 14. **c** Representative images for H&E- and trichrome-stained lungs of GF, ABSL-1 and ABSL-2 mice, 14 days after bleomycin injury with Scale bar, 100 µm. **d** Ashcroft Scoring of trichrome-stained lungs by housing conditions following bleomycin administration to mice. **e** Soluble collagen content of RLL of mice exposed to bleomycin or saline treatment and housed in gnotobiotic, ABSL-1, or ABSL-2 facilities by Sircol assay; Data are reported as mean ± SD with each dot representing an individual mouse, $n = 6$-$32$ mice. The experiments were independently repeated an average of two times. Statistical significance was assessed using a two-way ANOVA with Tukey's multiple comparison's test. *$P < 0.05$ **$P < 0.01$, ***$P < 0.001$ ****$P < 0.0001$; GF: Germ-free, ABSL-1: Animal Biosafety Level 1, ABSL-2: Animal Biosafety Level 2, ns = no significance; RLL: right lower lobe.

analysis was conducted using both the Jaccard index (Fig. 2c) and Bray-Curtis dissimilarity metric (Fig. S3) that account for presence/absence of taxa and taxon abundance variation, respectively (PERMANOVA, Jaccard: $F_{2,47} = 4.735$, $R^2 = 0.090$, $p = 0.003$; Bray-Curtis: $F_{2,47} = 4.676$, $R^2 = 0.089$, $p = 0.006$). Analysis of the gut microbiome of saline and bleomycin-treated mice revealed

distinctions in their microbial community on the ABSL-1 floor but not the ABSL-2 mice. Bleomycin treatment was associated with gut microbiota taxonomic composition for mice on the ABSL1 floor, but only when examining Jaccard distances (Bray-Curtis: $F_{1,17} = 0.727$, $R^2 = 0.041$, $p = 0.507$; Jaccard: $F_{2,17} = 9.392$, $R^2 = 0.356$, $p < 0.001$), akin to prior reports[23,24]. Distinctions by

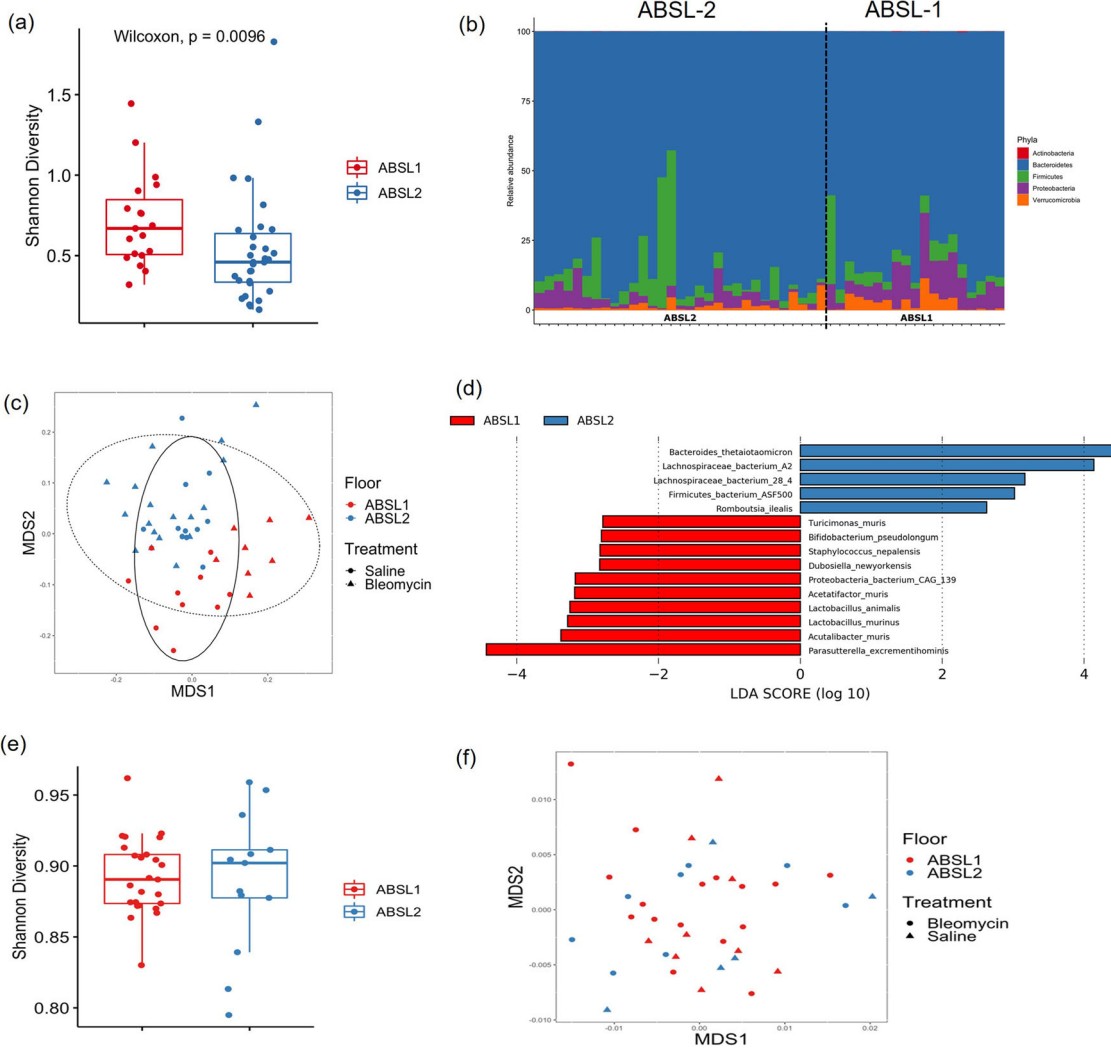

**Fig. 2 Increased gut microbial diversity is associated with better outcomes in murine model of fibrosis. a** Shannon diversity index scores for mice housed in ABSL-1 and ABSL-2 facilities and intranasally inoculated with 75 µl containing 0.04 Units of bleomycin or saline (saline) ($n = 13$-27 mice per cohort). The boxes show the median, as well as the 25$^{th}$ and 75$^{th}$ quartiles. The whiskers extend to 1.5*IQR. Each dox represents one mouse. **b** Phyla-level composition of the gut microbiome of ABSL-1 and ABSL-2 mice. **c** Nonlinear multidimensional scaling (MDS) plot showing differences in microbial taxonomic composition based on Jaccard dissimilarities. **d** Results of linear discriminant analysis showing taxa that are overrepresented in ABSL-1 or ABSL-2 mice. **e** Within-sample diversity for lung microbiome samples from mice housed on ABSL1 and ABSL2 floors. **f** Nonmetric multidimensional scaling plot based on Bray-Curtis dissimilarities showing the lung microbiome communities of mice housed on ABSL1 and ABSL2 floors. The experiment was independently conducted once.

treatment was not associated with taxonomic composition on the ABSL2 floor (Bray-Curtis: $F_{1,29} = 0.577$, $R^2 = 0.020$, $p = 0.547$; Jaccard: $F_{2,29} = 0.807$, $R^2 = 0.027$, $p = 0.561$) (Fig. 2c).

Using linear discriminant analysis (LDA) to examine species-level differences in the gut microbiota, 10 taxa were overrepresented in ABSL-1 mice and 5 taxa were overrepresented in ABSL-2 mice (Fig. 2d). The overrepresented taxa in ABSL-2 mice included *Lachnospiraceae bacterium A2*, *Lachnospiraceae bacterium 28-4*, *Firmicutes bacterium ASF500*, and *Romboutsia ilealis*. Higher relative abundance of Firmicutes in the lung microbiota of bleomycin-treated mice with fibrosis has been reported[7]. Species overrepresented in ABSL-1 mice included *Staphylococcus nepalensis*, *Dubosiella newyorkensis*, *Acetatifactor muris*, *Lactobacillus animalis*, *Lactobacillus murinus*, and *Acutalibacter muris*. Of note, *Lactobacillus* species, which were overrepresented in ABSL-1 mice, have been shown to reduce IL-17A-mediated disease severity in other experimental animal models[25–27], and were associated with notable reductions in respiratory symptoms[28].

**Gut, not lung, microbiome diversity is present between ABSL-1 and ABSL-2-housed mice.** Within-sample diversity of the lung microbiome does not differ significantly between floors (ABSL1 vs. ABSL2) or based on treatment (saline vs. bleomycin) when examined using either Shannon diversity (Floor: $W = 137$, $p = 0.447$; Treatment: $W = 196$, $p = 0.410$) or Shannon evenness (Floor: $W = 137$, $p = 0.447$; Treatment: $W = 214$, $p = 0.171$) (Fig. 2e). The composition of the lung microbiome of mice does not differ significantly between floors (ABSL1 vs. ABSL2) or based on treatment (saline vs. bleomycin) when examined using either Bray-Curtis dissimilarities (Floor: $F_{1,37} = 1.041$, $R^2 = 0.028$, $p = 0.388$; Treatment: $F_{1,37} = 0.993$, $R^2 = 0.027$, $p = 0.397$) or the Jaccard index (Floor: $F_{1,37} = 1.058$, $R^2 = 0.029$, $p = 0.343$; Treatment: $F_{1,37} = 0.733$, $R^2 = 0.020$, $p = 0.948$) (Fig. 2f).

Beta diversity differences in metagenomic functional gene profiles between rearing environment and treatment conditions were also examined using Jaccard distances (Figs. 3a and 3b) and Bray-Curtis dissimilarities (Fig. S4a and S4b). Functional profiles

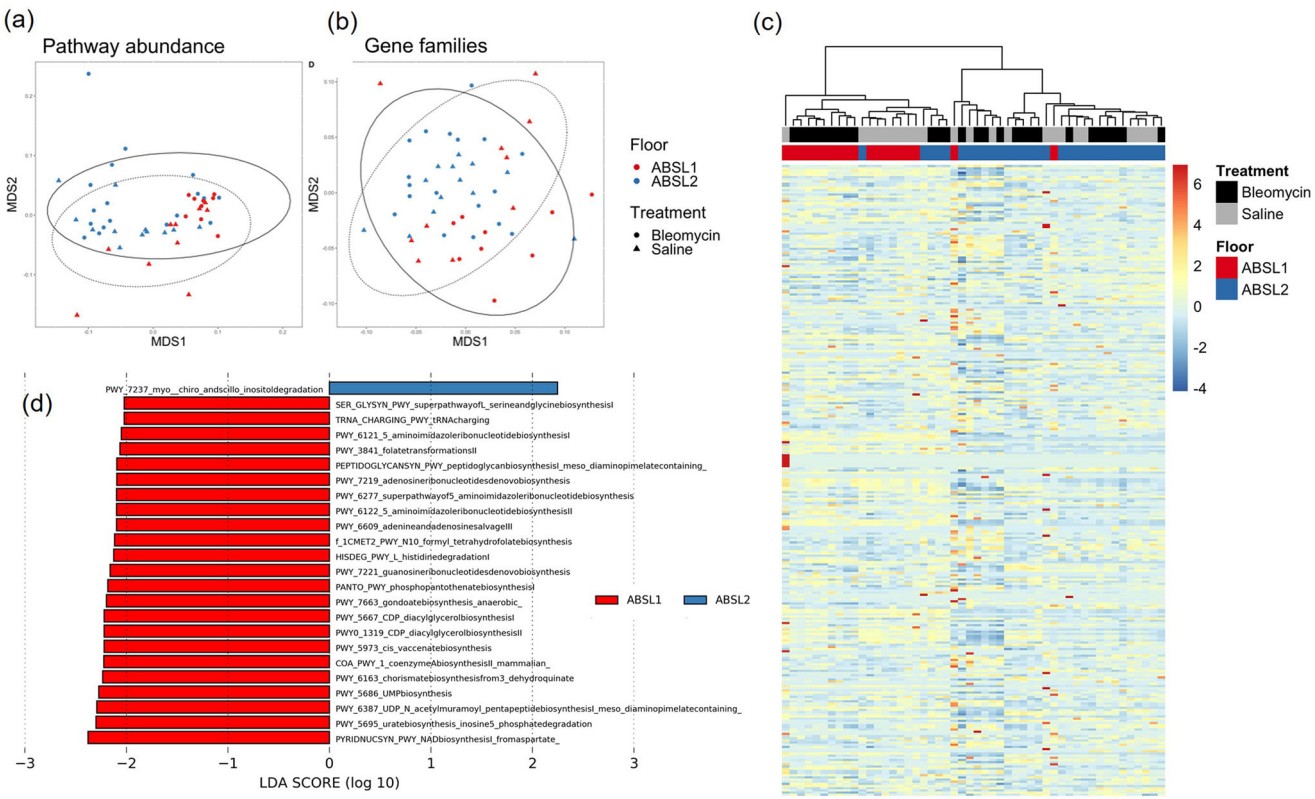

**Fig. 3 Functional gene profiles are significantly associated with rearing environment and treatment. a** MetaCyc reaction pathway abundance and **b** gene family abundance profiles using nonlinear multidimensional scaling (MDS) plots based on Jaccard distance matrices. **c** Heatmap showing normalized abundances of MetaCyc reaction pathway abundances between rearing environment and treatments. **d** Results of linear discriminant analysis showing MetaCyc reaction pathways that are overrepresented in ABSL1 or ABSL2 mice). $n = 13$-27 mice per cohort. The experiment was independently conducted once.

across gene family and MetaCyc reaction pathway abundances were significantly associated with environment and treatment specifically when examining Jaccard distances (Fig. 3c). MetaCyc reaction pathway abundances differed significantly between the gut microbiota composition of ABSL-1 and ABSL-2 mice (Jaccard: $F_{2,47} = 3.055$, $R^2 = 0.058$, $p < 0.001$) (Fig. 3c). Similarly, rearing environment was significantly associated with gene family profiles (PERMANOVA; Jaccard: $F_{2,47} = 7.030$, $R^2 = 0.125$, $p < 0.001$) (Fig. 3b). A difference between rearing environments only when using Jaccard distances, and not Bray-Curtis dissimilarities, indicated that low abundance pathways are driving the differences between rearing environments. Shannon diversity was higher in ABSL-1 mice for both MetaCyc reaction pathways ($W = 544$, $p < 0.001$) and gene families ($W = 149$, $p = 0.003$), but did not differ between saline and bleomycin treatments. GLMs did not identify any differentially abundant pathways between ABSL-1 or ABSL-2 mice. However, LDA identified 24 pathways that were over-represented in ABSL-1 mice and one pathway that was over-represented in ABSL-2 mice (Fig. 3d). Microbial community compositional and functional differences between mice in separate rearing groups with differing amount of fibrosis support the hypothesis that the gut microbiota may directly contribute to the degree of lung fibrosis. Metagenomic analysis on fecal pellets from mice in each housing cohort experiments was conducted once.

**Gut microbial diversity and taxa associate with proinflammatory lung responses**. To determine the immunologic drivers of lung fibrosis under distinct rearing environments, we performed flow cytometric analysis of single cell lung suspensions obtained from GF, ABSL-1, and ABSL-2 mice 14 days after inoculation with either saline or bleomycin. We noted a distinctive increase in

CD4 + IL6 + T cells with bleomycin treatment, in ABSL-2 mice compared to GF or ABSL-1 mice (Fig. 4a; representative histogram in Fig. S5a). The same trend occurred for the IL-6 co-receptor, gp130, where bleomycin-treated mice in the ABSL-2 cohort had the highest percentage of CD4 + gp130 + T cells (Fig. 4b representative histograms in Fig. S5b). We previously reported that PD-1 + Th17 cells contribute to the development of lung fibrosis. We did note that ABSL-2 mice treated with bleomycin had higher levels of PD-1 + CD4 + T cells secreting IL-17A than ABSL-1 or GF mice (Fig. 4c). Representative histograms and gating strategy are shown (Fig. 4d, e, S5c). Phosphorylated STAT3 (pSTAT3Y[705]) indicates that the transcription factor has been activated and can induce downstream inflammatory signaling to promote Th17 cell maturation. Bleomycin treatment increased pSTAT3 expression in CD4 + T cells in the ABSL-2 cohort mice (Fig. 4f). PCR analysis for *STAT3* mRNA demonstrated the same trend (Fig. 4g, S5d). There was a notable difference between ABSL-2 bleomycin and saline treated cohorts in the expression of IL-23R (Fig. S6a). The distinctions in the percentages of CD4 + IL-6+ and CD4 + IL-17A + T cells among mice in different housing cohorts support microbiota-induced alterations to the IL-6/ STAT3/IL-17A pathway in pulmonary fibrosis.

As there were differences between immune variables in the mice with different microbiota, we looked for associations between individual taxa abundances and immunological variables using Spearman correlation tests. Several taxa, including members of the Firmicutes and Proteobacteria, were found to be negatively correlated with multiple immunological values, including levels of CD4 + PD-1+ IL-17A + T cells and CD4 + pSTAT3 + T cells (Table 1). *Romboutsia ilealis* was positively correlated with those immunological values (Table 1). No microbial taxa were correlated either positively

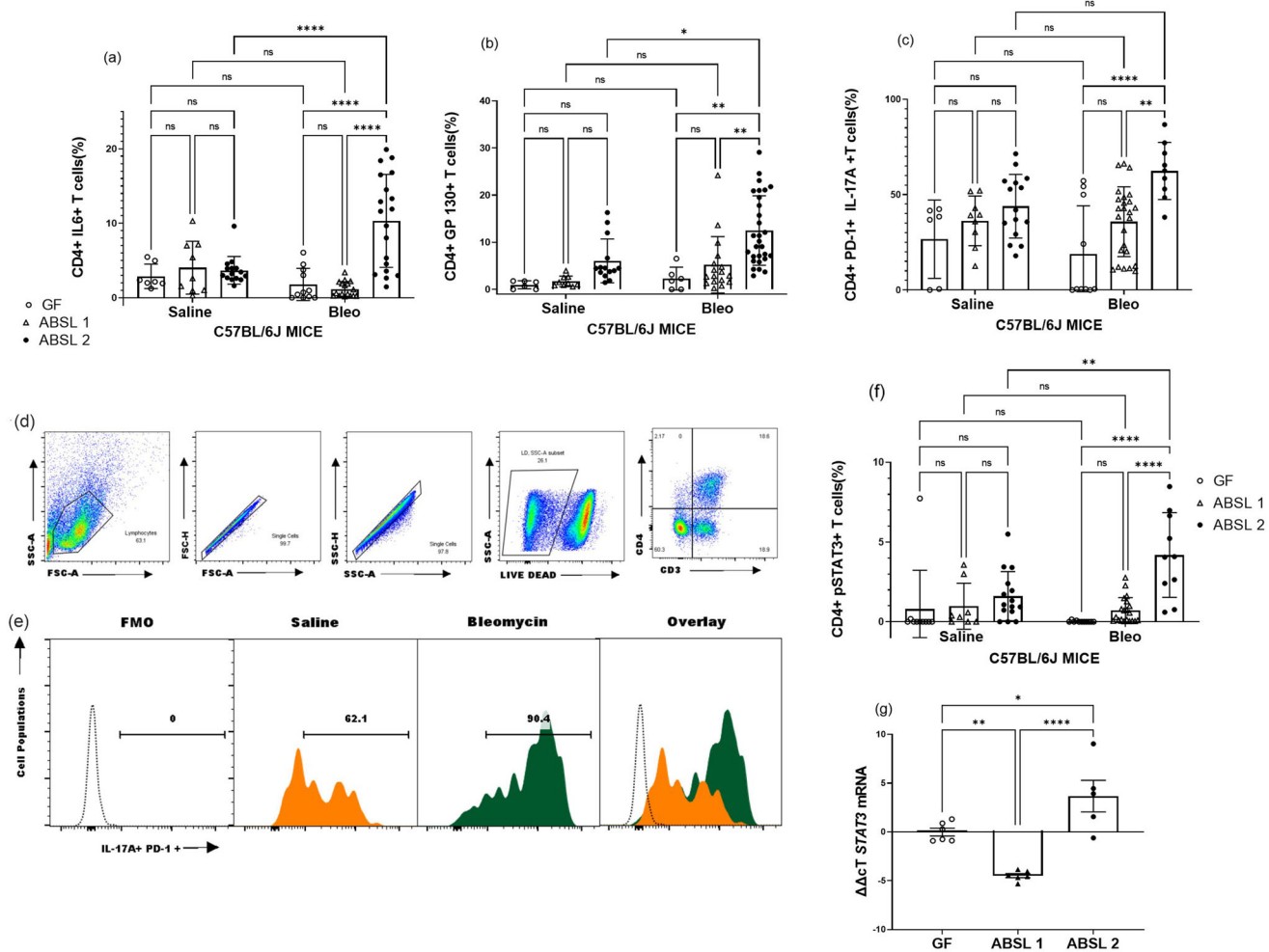

**Fig. 4 Germ-free mice do not mount a proinflammatory response upon bleomycin treatment.** Mice were housed in germfree, ABSL-1, or ABSL-2 facilities and intranasally inoculated with 75 µl containing 0.04 Units of bleomycin nor saline (control). 14 days postinoculation, mice were sacrificed and single-cell lung suspension prepared form the lungs. Flow cytometric analysis of T cells from single cell were conducted for CD4 + **a** IL-6, **b** gp130, **c** PD-1+ IL-17A + , **d** gating strategy, **e** representative histograms from ABSL 2 cohort and **f** pSTAT3Y[705] expression on T cells. **g** Relative expression of *STAT3* mRNA in lung tissue. Comparisons between cohorts were performed using two-way ANOVA with Tukey's post-hoc. *$P < 0.05$, **$P < 0.01$, ***$P < 0.001$, ****$P < 0.0001$, ns: no significance. Bars are mean ± SD; each symbol represents an individual mouse, $n = 6$–29 mice. The experiments were independently conducted an average of two times.

**Table 1 Spearman correlation associations between individual taxa abundances and immunological variables.**

| | CD4 + PD-1 + IL-17A + T cells | CD4 + pSTAT3 + T cells |
|---|---|---|
| *Dubosiella newyorkensis* (Firmicutes) | $\rho = -0.548$, $p_{fdr} = 0.011$ | NR |
| *Lactobacillus animalis* (Firmicutes) | NR | $\rho = -0.619$, $p_{fdr} = 0.012$ |
| *Lactobacillus murinus* (Firmicutes) | NR | $\rho = -0.619$, $p_{fdr} = 0.001$ |
| *Parasutterella excrementihominis* (Proteobacteria) | NR | NR |
| Proteobacteria bacterium CAG.139 (Proteobacteria) | NR | NR |
| *Romboutsia ilealis* (Firmicutes) | $\rho = 0.412$, $p_{fdr} = 0.043$ | $\rho = 0.599$, $p_{fdr} = 0.012$ |
| *Staphylococcus nepalensis* (Firmicutes) | $\rho = -0.407$, $p_{fdr} = 0.044$ | $\rho = -0.608$, $p_{fdr} = 0.012$ |
| *Turicimonas muris* (Proteobacteria) | NR | NR |
| Spearman correlation coefficients and FDR-corrected *p*-values are reported. NR = no significant relationship after FDR correction. | | |

or negatively with CD4 + IL-6 + T cell levels. The species negatively correlated with immune activation include *Lactobacillus* species, which have been shown to produce butyrate that dampens Th17 polarization[18,19].

**Reconstitution of low-diversity gut microbiota enhances pulmonary fibrosis.** To test the hypothesis that the gut microbiota causes fibrosis severity, we performed fecal microbiota transplantation (FMT) experiments. GF mice were gavaged every other day with ABSL-1 or ABSL-2 stool for one week prior to bleomycin inoculation and during the 14 days following the induction of lung injury. There were no survival distinctions between untreated GF mice, or GF mice gavaged with supernatant of ABSL-1 or ABSL-2 stool slurry (Fig. 5a). We observed a trend in increased weight loss at day 14 among GF mice treated with ABSL-2 stool, compared to untreated GF mice or GF mice

gavaged with ABSL-1 stool (Fig. 5b). Representative images for Masson's trichrome-stained lung histologic sections (with Scale bar) of GF, GF + ABSL-1 and GF + ABSL-2 mice are shown 14 days after bleomycin intranasal inoculation (Fig. 5c). We noted a marked increase in lung collagen content in the GF mice that underwent FMT of ABSL-2 stool. No increase in lung fibrosis was detected among GF mice following FMT of ABSL-1 stool (Fig. 5d). Flow cytometric analysis of the gavage mice revealed higher CD4 + T cells in the single cell lung suspensions of GF mice gavaged with ABSL-2 stool, compared to ABSL-1 or GF stool (Fig. 5e). Representative pseudo color plot of CD4 + T cells is shown (Fig. 5f). This data suggests that lung injury alone is not sufficient to induce fibrosis but that bacteria capable of inducing profibrotic cytokine expression also need to be present in the gut.

**Lactobacilli present in ABSL-1 stool samples reduces Human Lung Fibroblast (HLF) collagen production.** Analysis of the gut microbiome of bleomycin-treated mice revealed higher microbial diversity in the mice housed in the ABSL-1 floor. These mice also had reduced lung fibrosis and fewer Th17 cells. One of the ABSL-1 microorganisms identified in greater abundance was lactobacilli, which has been reported to reduce IL-17A expression. We examined the impact of supernatant from *Lactobacillus rhamnosus* in logarithmic phase of growth on Human Lung Fibroblast *(HLF)* collagen production, stimulated by the profibrotic cytokines, IL-17A and TGFβ1. We noted that serum-starved HLF from five different individuals demonstrated increased col1A production following the addition of IL-17A and TGFβ1($P < 0.05$, unpaired two-tailed $t$ test) (Fig. 6a). No increase in HLF col1A production was noted with the addition of *Lactobacillus* supernatant (Fig. 6b). *Lactobacillus* supernatant did significantly inhibit HLF collagen production induced by IL-17A and TGFβ1 controls ($P < 0.05$, unpaired two-tailed $t$ test) (Fig. 6c). We also noted a decrease in col1A by addition of sodium butyrate, although not significant ($P = 0.07$, unpaired two-tailed $t$ test) (Fig. 6d). Immunoblot analysis for col1A on the cell lysates confirmed these findings (Fig. 6e). Quantification of immunoblot analysis of Col1A is shown in Fig. 6f. These findings demonstrate the capacity of *Lactobacillus* to inhibit HLF collagen production.

## Discussion

Using the murine model of pulmonary fibrosis under distinct rearing environments, we establish the impact of variation in gut microbial diversity on lung fibrosis severity. Particularly, low gut microbial diversity was present among ABSL-2 mice that possessed severe lung fibrosis, whereas higher microbial diversity was present among ABSL-1 mice with less lung fibrosis following bleomycin administration (Fig. 2). Despite intranasal bleomycin treatment, GF mice did not develop lung fibrosis. FMT of ABSL-2 stool into GF mice before bleomycin treatment enhanced fibrosis severity, directly linking gut microbial composition with disease outcomes (Fig. 5).

We observed protection from bleomycin-induced acute lung injury in GF mice (Fig. 1), supporting the hypothesis that acute lung injury alone is not sufficient to induce lung pathology, but rather "two-hits" are necessary to induce pulmonary fibrosis: 1) acute lung injury (ALI) and 2) presence of microorganisms capable of inducing profibrotic cytokine expression, such as IL-17A. Remarkably, GF mice demonstrate minimal collagen deposition following intranasal bleomycin instillation. The significance of the gut microbiota to lung fibrosis progression was further supported by FMT of ABSL-2 stool into GF mice to recapitulate fibrotic lung disease severity (Fig. 5). FMT of ABSL-1 stool into GF mice did not induce notable lung fibrosis. Remarkable distinctions in lung microbial diversity were not detected in the bleomycin-treated murine model

(Fig. 2). This is distinct from publications of the lung microbiome in humans with lung fibrosis. Patients with fibrotic lung disease have exposure to multiple factors which can alter their lung microbiome, such as tobacco smoke and antibiotics[29,30]. Human are also mobile, and thus have exposure to multiple distinct environments, which can alter the lung microbiome. A concomitant analysis of the gut and lung microbiome in patients with lung fibrosis would be interesting. A recent clinical trial noted reductions in the incidence of upper respiratory tract infections among patients randomized to probiotics containing *Lactobacillus* and *Bifidobacteria* compared to placebo[28], further supporting the importance of the gut-lung axis in lung disease.

Investigation of the pulmonary inflammatory milieu among the different cohorts of mice provides an interesting linkage between profibrotic cytokine expression and reduced microbial diversity. It has been noted that dysregulation of lung microbiota increases IL-17B expression which drives pulmonary fibrosis[23]. Independent investigators support the capacity of the gut microbiota to induce systemic inflammation, specifically increased IL-6, IFN-γ and IL-17A, through mechanisms such as 1) formation of IsoLG-protein adducts in dendritic cells, which drive IFN-γ and IL-17A production by T cells;[31] 2) augmentation of gut permeability allows translocation of bacterial endotoxin (lipopolysaccharide, LPS) from the gut microbiota to blood circulation, leading to microbiota-dependent interleukin-17A (IL-17A) secretion from lamina propria derived Th17 cells;[32] or 3) reduced expression of short-chain fatty acids, such as butyrate, by the gut microbiota enhances Th17 cell expansion[33]. It is also noteworthy that microbial strains shown to improve gut permeability, such as *Bifidobacterium*, were most prevalent in the ABSL-1 stool (Fig. 2). Importantly, many of the organisms enriched in the ABSL-1 cohort, including *Dubosiella newyorkensis, Staphylococcus nepalensis*, and two *Lactobacillus* species were also correlated with decreased immune activation (Table 1). The only species that had a positive correlation with immune activation, *Romboutsia ilealis*, was enriched in the ABSL-2 mice. *Romboutsia* species are associated with the metabolic changes associated with obesity[34], which serves as an independent risk factor for sarcoidosis pulmonary progression[35]. Further investigation on the impact of specific microbial species, such as *Romboutsia* on metabolic disorders associated with lung fibrosis is warranted. Metagenomic analysis also revealed relatively higher abundance of *Lactobacillus* species in ABSL-1 housed mice (Fig. 2), which demonstrated fewer Th17 cells (Fig. 4). Various *Lactobacilli* species can reduce IL-17A expression via TLR2/4 in epithelial cells[27] or through the production of metabolites such as butyrate[25]. SCFA, like butyrate, reduce inflammatory conditions following insult through their capacity to decrease IL-17A synthesis, as well as promote IL-10 generation through T regulatory cell development[19]. We noted remarkable reductions in HLF collagen production induced by IL-17A and TGF-β1 (Fig. 6). Modulation of the gut microbiota of mice by oral gavage of *Lactobacillus reuteri* protected mice from multiple sclerosis-like disease symptoms by decreasing IL-17A levels[26]. Future investigation of the impact of microbial diversity on lung fibrosis is warranted, including assessing the ability of probiotic treatment with *Lactobacillus spp.* to reduce disease severity in patients with lung fibrosis.

In addition to identifying members of the gut microbiota that may contribute to or protect from severe lung fibrosis, our investigation suggests that increased diversity in gut community composition is beneficial with respect to pulmonary disease severity (Figs. 1, 5). This observation follows with the ecological principle that more diverse ecosystems are more resistant to perturbations than less diverse systems. Diversity, rather than specific microbes, has been shown to be beneficial for both community stability and functionality over time, whether members of the ecosystem are animals, plants, or microorganisms[36–39]. Additionally, murine

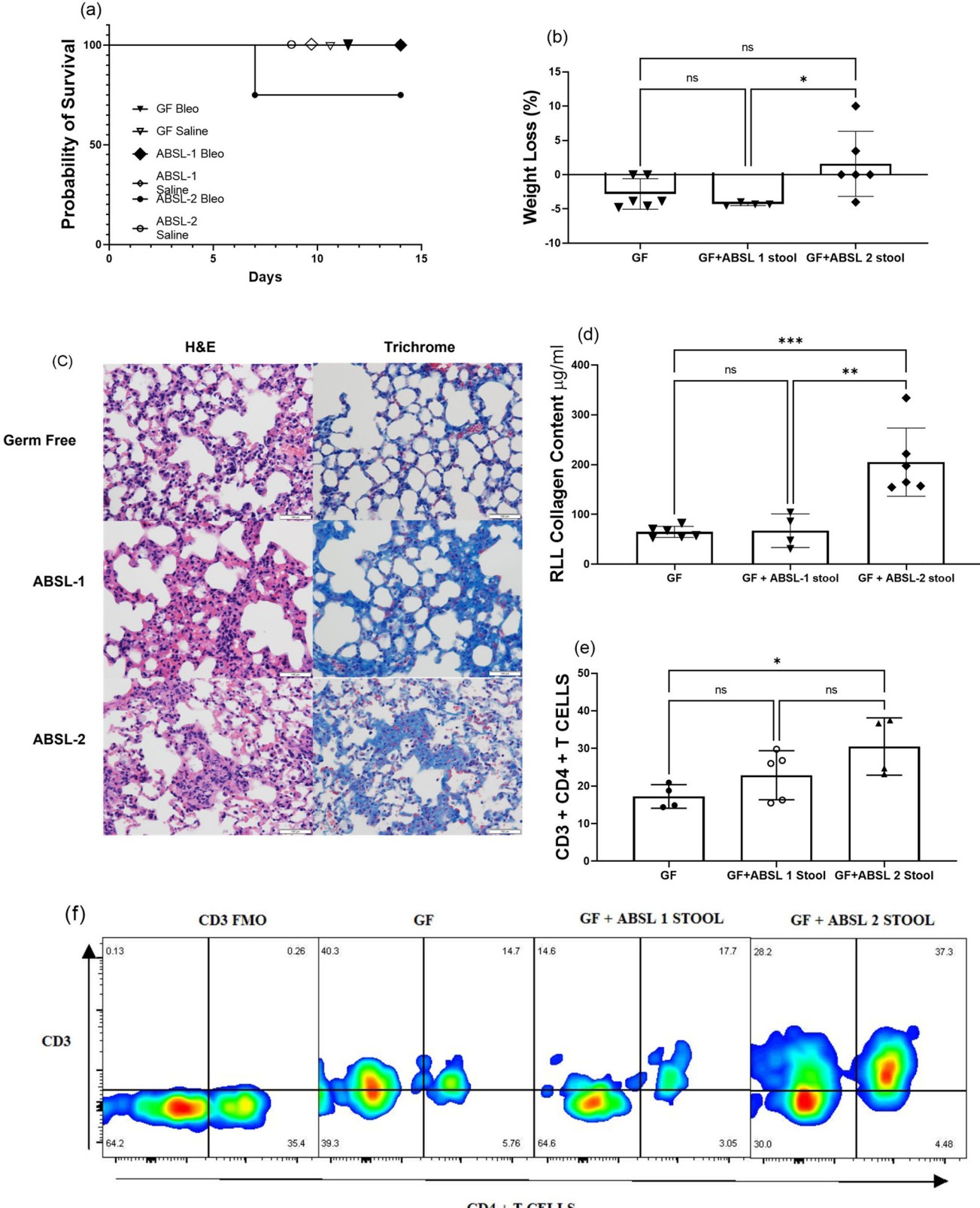

studies have shown that early-life perturbations to microbial gut community diversity can worsen inflammatory conditions such as allergies, psoriasis, and gut inflammation later in life[40,41]. The increased diversity in the gut microbiome of the ABSL-1 cohort mice might be playing a protective role in promoting healthy and controlled immune responses (Fig. 7).

The human gut microbiota represents a thriving ecosystem with taxonomic and functional diversity within and between individuals[42]. Fecal microbiota transplantation (FMT) has been explored clinically to increase gut microbial diversity to combat complications such as those occurring after allogeneic hematopoietic stem cell transplantation[43] and Clostridium difficile infection[44].

**Fig. 5 Reconstitution of an ABSL-2 microbiome in GF mice restores lung fibrosis morbidity.** Germ-free mice were gavaged with ABSL-1 or ABSL-2 fecal slurries and given bleomycin to induce lung fibrosis. **a** Survival curve of GF, GF + ABSL-1 and GF + ABSL-2 mice, 14 days after bleomycin injury. **b** Weight loss at day 14. **c** Representative images for Masson's trichrome-stained lungs of GF, GF + ABSL-1 and GF + ABSL-2 mice, 14 days after bleomycin injury with Scale bar, 100 μm. **d** RLL collagen content by Sircol assay were determined across the gavage treatment groups. **e** Single-cell lung suspensions were obtained from mice that were GF or gavaged with stool from mice housed in ABSL-1, or ABSL-2 facilities (GF + ABSL-1 and GF + ABSL-2). Flow cytometry analysis was conducted for CD4 + T cells and **f** representative pseudocolor plot of CD4 + T cells. Each symbol represents an individual mouse. Statistical significance was assessed using a one-way ANOVA with Sidak's multiple comparison's test. *P < 0.05, **P < 0.01, ***P < 0.001, ns = no significance n = 4-6 mice. The experiment was independently conducted twice.

Additional murine studies using FMT have demonstrated increased murine survival after acute liver injury and during acute respiratory distress syndrome by decreasing IL-17A levels[45,46]. This investigation adds to the growing body of FMT literature, suggesting that functional and taxonomic diversity in the gut microbiota of patients with lung fibrosis may improve their clinical outcomes, and that decreased gut microbial diversity may predispose individuals to more severe manifestations of fibrotic lung disease (Figs. 1, 5). Molecular examination of ABSL-1 stool demonstrated that there were some proinflammatory bacteria present, such as proteobacteria which has been linked with both inflammatory gut and airway environments in asthma[21] and after lung transplantation;[22] however, companion microorganisms, may limit their capacity to induce severe lung disease.

There are limitations to this investigation. While our sequencing studies focused on the bacterial members of the gut community, the potential role of fungal and viral contributors to the gut microbiota's impact on pulmonary fibrosis cannot be discounted. In addition, other components of stool, including host-derived cells and metabolites, are transferred to the naïve host during FMT[47]. Also, this study did not probe lung fungal and viral communities among mice in the different rearing environments; altered lung microbiota have been noted to be important in ILD patients[7,48]. Further investigation of the influence of the gut microbiota on the lung microbial community and vice versa is warranted.

Overall, our study implicates the low-diversity gut microbiota as a contributor of pulmonary fibrosis severity through their capacity to activate the IL-6/STAT3/IL-17A signaling pathway. These findings support that acute lung injury in a host with low microbial diversity leads to more severe lung disease, compared to hosts with increased microbial diversity. Validation of the potential benefit of enhancing gut microbial diversity in ILD patients is warranted.

## Methods

**Murine model of lung fibrosis.** The experiments in this study were designed to compare the impact of the housing environment, on gut microbial composition and the effect on lung fibrosis outcomes. To achieve this, we reared C57BL/6 J mice purchased from Jackson Labs or bred at the University of Louisville in either germ-free (gnotobiotic conditions with no microbial interactions, animal biosafety level 1 (ABSL-1) experimental conditions in which no infections were conducted, or animal biosafety level 2 (ABSL-2) environments in which infections with pathogenic microorganisms occur. None of the mice in any of the facilities underwent inoculation with any microorganisms; only bleomycin was administered intranasally to induce lung fibrosis. Mice were housed in groups of three per cage, and all mice in each cage received the same treatment. Randomization was not used to allocate experimental units to control and treatment groups. Determination of sample size was based upon a minimum of six mice per group. On average, there were 6-29 mice per group. Lower numbers of mice particularly in the germ-free group were due to the limited number of adaptive immune cells in the lungs of germ-free mice. Sacrificing additional germ-free mice would not guarantee enough immune events to conduct flow cytometric analysis. For the murine model of bleomycin-induced pulmonary fibrosis, 6- to 8-week-old C57Bl6 mice weighing approximately 17–22 g were used. Mice were anesthetized by intraperitoneal injection of 80 μl of 20 mg/ml Ketamine/1.8 mg/ml Xylazine solution, then 75 μl containing 0.04 Units of bleomycin (Novaplus Lake Forest IL) in saline (0.9% sodium chloride) (Hospira Inc, Lake forest IL) as control were administered intranasally to mice in each housing cohorts as reported[49]. No criteria were set for excluding animals from analysis a posteriori. A priori, we excluded any germ-free mice with evidence of infection by stool culture within 14 days of bleomycin administration. We also excluded any wild-type mice from the immune analysis and Sircol assay who died before 14 days, which is the timeframe required to develop lung fibrosis. Based on these criteria, we excluded six germ-free mice and six wild-type mice. Lungs were harvested for histology, flow cytometry, or single-cell isolation as reported[50]. The outcome measures were survival, weight loss, lung collagen quantification, immune analysis for IL-17A signaling pathway and microbiome analysis of stool and lung of mice by their distinct housing environment. The researchers performing outcome measures were blinded to the environment and treatment of each mouse. Each mouse was considered an individual data point in our analysis. All murine procedures were performed according to the protocol approved by the Institutional Animal Care and Use Committee at Vanderbilt University Medical Center (protocol #M1700043; ABSL-1 and ABSL-2 facilities, and FMT experiments) or University of Louisville (protocol #20786; Germ-free facilities) using C57Bl/6 male and female mice at 6-8 weeks of age. Germ-free isolators at the University of Louisville are sampled (swabs of the environment plus fresh fecal pellets from at least 3-4 random cages) and cultured aerobically on Chocolate Agar and TSA w/ 5% Sheep Blood Agar at 37 °C and anaerobically on Anaerobic PEA Agar, LKV/BBE Agar (Laked Kanamycin Vancomycin/Bacteroides Bile Esculin) and Brucella Agar at 37 °C each time the isolator is opened (roughly once/week but opened as few times as possible). No microbial growth was detected in any isolator used in this study.

**Analysis of lung fibrosis.** Lung fibrosis was assessed in mice given intranasal saline or bleomycin both quantitatively and qualitatively. The right lower lobe of each lung was homogenized in acetic acid and collagen content measured using the Sircol assay (BioColor Ltd., Newton Abbey, UK) following the manufacturer's protocol. The right middle lobe of each lung was immediately placed in formalin for fixing overnight before being transferred to ethanol prior to embedding and sectioning for histology. Slide preparation was performed by the Vanderbilt Translational Pathology Shared Resource (TPSR). Slides were stained with hematoxylin and eosin (H&E), as well as trichrome blue by TPSR. Blinded slides were scored using the updated Ashcroft scoring guidelines[51].

**Metagenomic analysis of gut microbiota.** Fecal pellets and lung samples were collected from mice in each housing cohort and genomic DNA (gDNA) was extracted with the Qiagen DNAeasy extraction kit (Qiagen, Valencia, CA) according to the manufacturer's instructions. The gDNA concentration and quality were confirmed using the Bioanalyzer 2100 system (Agilent, Santa Clara, CA).

Paired end sequences were trimmed, quality-filtered, and host contaminants were removed in KneadData (v.0.7.6) using default values and the C57BL mouse reference database. Taxonomic composition of gut microbiome samples was profiled using MetaPhlAn (v.3.0.4)[52] and functional composition was profiled using HUMAnN3 (v3.0.0)[53]. Samples with less than 10 million sequence reads remaining after quality filtering and removing host reads were removed prior to downstream analysis. As lung microbiome samples had lower percentages of sequences mapping to bacteria, we additionally used Kaiju (v.1.8)[3] with the nr_euk 2021-02-24 database to taxonomically classify both gut microbiome and lung microbiome samples for all analyses of the lung microbiome. Sequence of lung and gut microbiome have been deposited into BioProject ID PRJNA899808.

Permutational multivariate analysis of variance (PERMANOVA) was used to assess the effects of both housing (ABSL-1 vs. ABSL-2) and treatment (saline vs. bleomycin) on Bray-Curtis dissimilarity and Jaccard distance matrices constructed from species-level relative abundance tables. PERMANOVAs were performed using the *vegan* package in R[54,55]. Wilcoxon Rank Sum tests in R were used to examine differences in Shannon diversity and evenness between the ABSL-1 and ABSL-2 environments. We used linear discriminant analysis in LEfSe[56] to identify differentially abundant taxa and pathways across both housing and treatment groups. Generalized linear mixed effects models (GLMMs) using a negative binomial distribution using the *glmmTMB* package in R[57] were additionally used to identify differentially abundant taxa and functional pathways. Spearman correlations were used to examine associations between individual taxa abundances and physiological variables. Core microbiome analysis was used to identify species present in >70% of samples within a given group (gut microbiome from ABSL1 mice, lung microbiome from ABSL1 mice, gut microbiome from ABSL2 mice, lung microbiome from ABSL2 mice, gut microbiome from germ-free mice gavaged with ABSL1 stool, and gut microbiome from germ-free mice gavaged with ABSL2 stool,

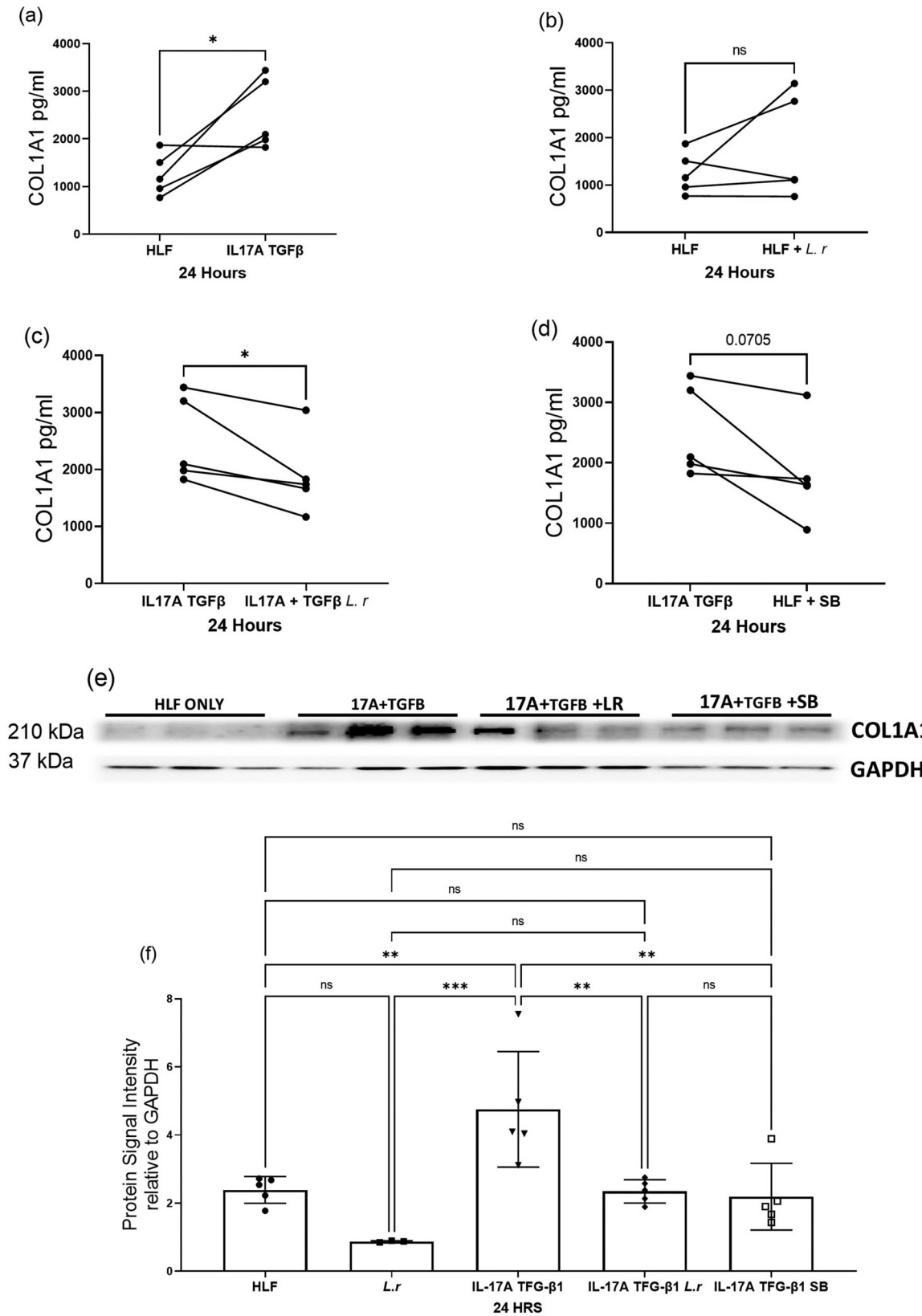

and lung microbiome from germ-free mice gavaged with ABSL2 stool). Overlap between core taxa between treatments was used to assess overlap between stool and lung microbiomes, as well as the success of the fecal transplant. False discovery rate (FDR) corrected p-values were calculated using the *fdrtool* package in R[58] and are reported for all GLMMs and Spearman correlations. Code for all analyses can be found at github.com/emallott/PulmonaryFibrosisMicrobiota.

**Flow cytometry**. All flow cytometry experiments were acquired with an LSR-II flow cytometer (BD Biosciences), and information on all antibodies used in this study is listed in Supplemental Table 1. Live cells were gated based on forward and side scatter properties, and surface staining of cells was performed as reported[59]. Th17 cells were identified using flow cytometry as previously described[60]. Cells were gated on singlets, live CD3 + and CD4 + cells. Data analysis was performed

**Fig. 6 *Lactobaccilli* abundant in ABSL-1 Stool Samples Reduces HLF Collagen Production.** HLF from five different individuals were serum starved in 0.4% FBS, then assayed for collagen type 1 A (col1a) production. Collagen type 1a production after HLF's are treated with **a** rIL-17A and TGFβ1, **b** supernatant from *Lactobacillus* grown to log phase, **c** supernatant from Lactobacillus grown to log phase and rIL-17A + TGFβ1 or **d** sodium butyrate. **e** Immunoblot of HLF samples with antibodies against collagen 1a and GAPDH, **f** quantification of immunoblot analysis. Each symbol represents an individual. Statistical significance was assessed using students t test. *$P < 0.05$, ns = no significance. n = 5 HLF's. Experiment was carried out using five different HLF cell lines. Each individual cell line was independently tested in triplicate, then the experiment was repeated once. The average of the individual cell lines in each experiment was used to generate the figure.

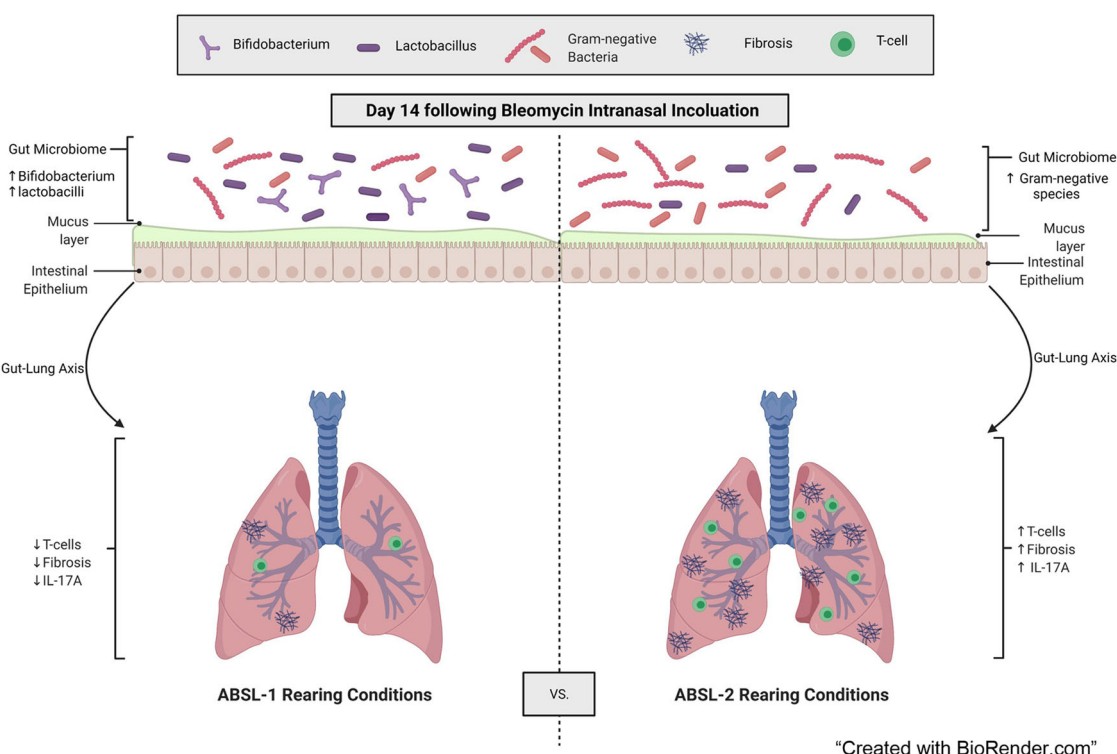

**Fig. 7 Model of gut contribution to fibrotic lung disease.** Direct modulation of the gut microbiota in a bleomycin-induced murine model suggests that a functional gut-lung axis attenuates lung fibrosis.

using FlowJo software (Tree Star, Ashland, OR). A minimum of 50,000 events were acquired per sample.

**Fecal microbiota transplantation experiments.** Fecal pellets were collected from mice housed in the ABSL-1 and ABSL-2 facilities. These pellets were suspended in sterile PBS to create fecal slurries at 100 mg feces/ mL PBS. Germ-free mice were given 100 μL of the supernatant of the fecal slurry by oral gavage every other day, beginning one week before intranasal inoculation with bleomycin. The germ-free mice gavaged with stool were also fed non-sterile food that originated from either the ABSL-1 or ABSL-2 facility, to match the stool received. Mice were humanely euthanized, and tissues collected 14 days after bleomycin treatment.

*RNA extractions and quantitative real time PCR analysis.* RNA was extracted using the Rneasy Mini kit (Qiagen, Hilden Germany). Subsequently 100 ng of RNA was converted to cDNA using the iScript™ cDNA Synthesis Kit (Biorad USA). Taqman quantitative PCR analysis of *Stat3 was* conducted and data was reported as relative expression normalized to the *Gapdh* housekeeping gene.

*Bacterial preparation and inoculation. Lactobacillus rhamnosus* (a gift from Microbiology lab at VUMC) was inoculated in MRS Agar and grown in MRS broth at 37 °C in 5% $CO_2$ overnight without shaking. Bacteria was grown to log phase (1 × 10$^9$ colony-forming units) and centrifuged at 3700 rpm for 7 minutes. The resulting supernatant was filtered in 0.22 μm filter and 100 μL of this was used in the HLF coculture experiment described below.

*Human Lung Fibroblast (HLF) and* Lactobacilli *supernatant co-culture experiments.* Four Human Lung Fibroblasts (HLFs) cell lines were obtained as a kind gift from Dr. Alexander Gelbard Department of Otolaryngology - Head & Neck Surgery Vanderbilt University Medical Center. The fifth cell line was purchased from Sigma Aldrich #506-05 A (The HLF Demographics are provided in

Table 2). HLF's are grown to confluence in appropriate media and maintained in the Dulbecco's modified Eagle's medium supplemented with 10% fetal bovine serum (FBS), 100 U/mL penicillin, and 100 mg/mL streptomycin under 5% $CO_2$ at 37 °C.

HLF's are plated in 96-well plates at a density of 3,000 cells/well overnight after which HLF were quiesced by serum starvation in 0.4% fibroblast basal medium (FBM) for 24 h before the addition of 20 ng/ml of rIL-17A (317ILB050, RHIL-17A 50 UG) and 10 ng/ml of rTGF-β1 (240B002, rHTGF-β1 2 UG). HLF with rIL-17A and rTGF-β1 then received 100 μL of *Lactobacillus rhamnosus* supernatant or 1 mM of sodium butyrate (SB) (NC0718494, Sodium Butyrate 100 G) for 24, after which Col1a was analyzed from the supernatant using the Collagen 1 Alpha 1 Kit from Abcam (catalog # ab210966). *Lactobacillus rhamnosus* was grown as described in the methods above, and 100 μL supernatant used for co-culture with HLF. HLF cell pellet was collected and used to analyze protein expression by ELISA or immunoblot analysis, respectively. Western blot analysis was conducted after a 24-hour incubation period. Cell culture supernatants were collected for either western blot or ELISA analysis.

*Lysates, SDS-PAGE, and western blotting.* Cells were lysed as reported[61]. In preparation for SDS-PAGE, HLF cell lysates were centrifuged for 15 min at 14,000 RPM at 4 °C. All cell lysates were adjusted for uniform loading by using the Bradford Protein Assay prior to gel loading and run. Cell lysates and supernatants were resolved by SDS-PAGE and then analyzed by western blotting with antibodies against Collagen-1 (Cell Signaling), and α-SMA (Abcam). Following gel electrophoresis, proteins were transferred onto Immobilon-FL PVDF membrane (Millipore Corporation). Stripping and blot incubations were performed as specified in Li-Cor western blot analysis protocol specific for the Odyssey Infrared Imaging System. Band visualization and densitometry was completed using Li-Cor Odyssey Infrared Imaging System (LI-COR Biosciences, Lincoln, NE) and studio software. For analysis, background intensity was subtracted from band intensity and the proteins of interest were normalized to GAPDH.

**Table 2 Human Lung Fibroblast Demographics.**

| Name | Information age, race, sex (tissue type) | IL17A + TGFβ-1 Collagen (pg/ml) | IL17A + TGFβ-1 + *L. rhamnosus* Collagen (pg/ml) | P Value |
|---|---|---|---|---|
| SKCO | 41 C F (HLF from subglottic scar)-patient with Idiopathic Subglottic Stenosis | 1823.053 | 1165.561 | 0.0003 |
| SNB | 52 C F (HLF from normal trachea mucous)-patient with Idiopathic Subglottic Stenosis | 3440.163 | 3037.827 | 0.0395 |
| NTH | 57 C F (HLF from normal trachea mucous)-patient with Idiopathic Subglottic Stenosis | 1981.196 | 1737.568 | 0.0020 |
| NNB | 52 C F (HLF from normal trachea mucous)-patient with Idiopathic Subglottic Stenosis | 3201.763 | 1826.211 | <0.0001 |
| Sigma Aldrich | #506-05A HLF Adult | 2095.598 | 1663.641 | 0.0067 |

*C* Caucasian, *F* Female

*Statistics and reproducibility.* Multiple-group comparisons were performed using a two-way analysis of variance (ANOVA) with Tukey's post hoc test. Statistical analysis for all figures was carried out using Prism version 7.02 (GraphPad Software). Data are reported as mean ± SD with each dot representing an individual mouse. All murine experiments were conducted at least twice resulting in final murine cohorts of 4-29 mice based on sample availability; the lung and gut microbiota analysis was performed once on a murine cohort comprised of 13–27 mice. A replicate was defined as an experiment conducted using at least nine wild-type mice and at least three germ-free mice. For a result to be considered statistically significant, a *P* value of less than 0.05 was used.

**Reporting summary.** Further information on research design is available in the Nature Portfolio Reporting Summary linked to this article.

## Data availability

All sequences obtained from the lung and gut microbiome analysis of germ-free, ABSL-1 and ABSL-2 mice has been deposited into BioProject ID, Accession number PRJNA899808. All data generated or analyzed during this study are included in this published article (Supplementary Data 1 file).

## Code availability

Code for all analyses can be found at github.com/emallott/PulmonaryFibrosisMicrobiota.

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

## Acknowledgements

This work was supported by grants NIH K24 HL127301-01, R56 RHL149129-01A1, and an Ellen Dreiling Research Fund Endowment to W.P.D and the Vanderbilt Microbiome Initiative. O.S.C is supported by T32 Grant T32AR059039-10, K12 HL 143956-4 and Foundation of Sarcoidosis Research FSR RFP 17-9041. The germ-free mice are bred and housed at the at University of Core facility supported by NIH/NIGMS CoBRE grant (P20GM125504). S.R.B, S.D and B.H are supported by P20GM1125504, as well as at Vanderbilt University School of Medicine Animal Facility. The model was created using BioRender.

## Author contributions

O.S.C. designed and performed experiments, analyzed data, and wrote the manuscript. E.K.M. performed the metagenomic analysis and wrote the manuscript. A.C., J.C.V., N.D., H.W., M.K., M.B.P., J.J., and B.S.G. performed experiments. S.R. Bordenstein and G.R.B. contributed to the intellectual content, study design and editing of the manuscript. S.D., S.R. Bodduluri, and H.B. aided with the germ-free experiments. W.P.D. conceived the study, designed experiments, analyzed data, and wrote the manuscript.

## Competing interests

The authors declare no competing interests.
