## [Peer Review File · Communications Biology]

Reviewers' comments:

Reviewer #1 (Remarks to the Author):

In this study, the authors investigated the bleomycin-induced murine model in different environment, and explore the role of gut microbiota in modulating lung fibrosis via gut-lung axis, which indicated the impact of variation in gut microbial diversity on lung fibrosis severity. However, some major concerns and limitations are still needed to be elucidated.

Major comments

1. Why do you compare the environment of animal biosafety level 1 and level 2?
2. As far as I know, it is still not clear how gut microbiota affect the lung fibrosis, but some studies have elucidated the mechanisms of how lung microbiota influence the pulmonary fibrosis (PMID: 30824326, PMID:30789747). Can you describe the relation between the gut microbiota and lung fibrosis and the difference of gut microbiota and lung microbiota in lung fibrosis.
3. The manuscript suggested that bleomycin treatment does not alter the diversity of the gut microbiota, it is different from the previous studies (PMID 33803282, 31473797). Therefore, can you elucidate the reasons.
4. In figure 4a,4b,4c, the number of mice models in some group is not enough to elucidate the conclusion.

Minor comments

1. What are the other indicators for evaluating pulmonary fibrosis?
2. The manuscript contains some typographical and language errors. You are advised to seek the assistance of a professional manuscript editing service.
3. The style of legends in figure 1 are not identical, such as (a),(b),(c),(d).
4. There are some other indicators of alpha diversity as well as beta diversity, are there any difference?

Reviewer #2 (Remarks to the Author):

1. Brief summary of the manuscript

The article entitled "Gut microbiota modulates lung fibrosis severity following acute lung injury" by Chioma et al. shows that rearing environment influences the response to intranasal bleomycin induced lung fibrosis. Germ-free mice were protected from lung fibrosis as well as mice reared in an environment providing a more diverse fecal microbiota. The study shows that in the mice with the less diverse microbiota, bleomycin administration induced more CD4+IL-17A+ T cells, which could be responsible for the more severe phenotype. Finally, after transplanting the highly diverse or the less diverse microbiota into germ-free recipient mice, bleomycin-induced lung injury reproduced some of the characteristic traits observed in the mice reared in the two distinct environments (providing highly or less diverse microbiota).

2. Overall impression of the work

This is a very interesting study providing new data about how gut microbiota can influence lung response to injury and fibrosis pathogenesis that will be of interest to others in the field. However, it would greatly benefit from additional data on the lung microbiota. Indeed, it is assumed that all the differential effects induced by the intranasal administration of bleomycin are caused by the differences in gut microbiota. It is very likely that the lung microbiota would also be different between the 3 mice groups (germ-free, ABSL-1 and ABSL-2) and could be partly responsible for the changes in response to bleomycin. The authors also performed FMT from ABSL-1 and ABSL-2 feces into GF recipient mice and recapitulate (partly) the phenotype observed in the original experiment. This suggests that the differences in phenotypes are due to changes in gut microbiota although there is no evidence showing that the microbes (or some of them) did not also colonize the lungs especially when oral gavages occurred every other for 3 weeks. I understand that these are extensive additional experiments but I believe these data would be necessary to be able to state that "the gut microbiota modulates lung fibrosis severity following acute lung injury".

3. Specific comments, with recommendations for addressing each comment

Major

1. Differential effect of gut and lung microbiota. It is not fully demonstrated that the changes in response to bleomycin are fully and only due to the differences in gut microbiota composition. This study would greatly benefit from including the composition of the lung microbiota in the different groups (GF, ABSL-1 and ABSL-2) and after FMT. Indeed, if fecal microbes were not able to colonize the lungs after FMT, this would strongly strengthen the role of the gut microbiota in the fibrosis pathogenesis.

2. Animal experiments: How were the number of animals per groups determined? According to the ARRIVE guidelines, the authors should "explain how the sample size was decided and provide details of any a priori sample size calculation, if done". This is especially important as the differences in group sizes are sometimes very important (2 to at least 25). If the authors believe that adding the n numbers on the figures would make them unreadable, this information could be included in the methods section. Please also include how many, if any, independent experiments were performed and which measurements/parameters were looked at for each experiment. Indeed, the n numbers also vary greatly within the same group when looking at different parameters (for example, on fig 4A, the GF saline group has 2 animals whereas it has 5 or 6 mice on Fig4B, ABSL-2 bleo group has 19 mice on fig 4A and >25 on Fig 4B...)

3. Impact of bleomycin on gut microbiota: It is somewhat surprising that bleomycin induced weight loss in ABSL-1 and ABSL-2 mice but did not seem to have an effect of gut microbiota composition. Intranasal bleomycin has been shown to alter lung microbiota (Yang et al Immunity 2019 PMID: 30824326) and weight loss/reduced food consumption usually induce changes in microbiota composition (Sencio et al Cell Rep 2020 PMID: 32130898)

4. Statistical analysis: Most experiments have 2 variables (rearing environment and saline/bleomycin administration). Therefore the statistical analysis should be done using a two-way ANOVA instead of a one-way ANOVA

5. FMT study: Was there any metagenomics study performed on feces from transplanted animals to confirm that alpha and beta diversity indexes were similar to that of ABSL-1 and ABSL-2 mice? This is very important to have an idea about how the microbiota implanted itself in the GF recipient mice.

Minor

Introduction

6. Page 2, line 19, Lactobacilli and Bifidobacteria are presented as butyrate producers. Although it is true that Lactobacilli do produce butyrate, Bifidobacteria produce mainly acetate and lactate and butyrate producers are mainly Bacteroidetes and Clostridia.

Results

Figure 1:

7. Panel A. The legend is very confusing, especially between the ABSL-1 saline and bleo groups: it looks like the ABSL-1 bleo mice survived better than the ABSL-1 saline mice. Could the authors change the symbols for these groups so that the figure is read better?

8. Panel B. The GF saline mice all show 0% change in weight, which is very surprising. How was the weight loss calculated? Was there any "normalization" done using the GF saline as the control group so that they would not show any weight change? It will be helpful to have this stated in the methods section.

9. Panel C. It is difficult to see the scale bars. Please include them on all images and make them a bit thicker to make them easier to see.

10. Panel E caption. It is stated that the number of mice is 6-25 but it is mentioned 2-19 in the reporting summary. Please make sure all the n numbers (for mice and independent experiments) are reported in the methods section.

Figure 2:

11. Panel A. Please state in the caption what the boxes are representing (mean/median/SD...)

12. Panel A/C. Does one dot represent one mouse?

13. Panel A/B/C. Please state the number of animals used in this experiment.

14. Figures 2/S2/3/S3. It is unclear why Bray-Curtis diagram is shown as beta-diversity for microbiota composition in the main figure whereas the Jaccard index diagram is shown for the beta-diversity of metagenomics functional genes. Could the authors clarify and/or homogenize which diagrams are part of the main figures and which ones are part of the supplementary files.

15. On page 6 lines 11 to 14, "ABSL-1 mice had significantly higher relative abundances of Proteobacteria and Verrucomicrobia. Increased proteobacteria abundance has been linked with both inflammatory gut and airway environments, as in Crohn's disease, asthma and after lung transplantation". This suggests that ABSL-1 mice would show more inflammation than ABSL-2 mice, which is not the case if we look at the results on IL-6/IL-17 further in the paper. Unfortunately this is not discussed although it is interesting. Do the authors think that the diversity rather than the composition itself is responsible for the differences in proinflammatory responses to bleomycin?

Fig S5

16. Panel A. It looks like saline induced an increase in the IL-6+ T cell population in ABSL-1 mice. Do the authors have an interpretation for this? Also, is it known why the FMO profiles are different between GF/ABSL-1 and ABSL-2?

17. Panel B. Looking at the overlay diagrams, saline and bleo seem to induce a bigger increase in GP130+ cells in ABSL-1 mice than ABSL-2 mice. Do the authors have an explanation for this?

18. Panels A/B. Selections of the positive cells, represented by the bars, seem shorter in ABSL-1 diagrams than in GF and ABSL-2 (which seem to be of similar sizes). Could the authors explain if this is a technical bias or if this has a scientific meaning?

19. On page 7, line 2-3: is there a particular reason for the italics on "overrepresented in ABSL-2 mice (Fig2d)"?

20. Figure S4. It is confusing why on panel A, there are only 2 mice in the CF saline groups and on panel B, there are only 2 mice in the GF Bleo group. Could the author clarify why the number of mice per group do not always match throughout the figures? The same is true for figure 4 panels a, b, c and f.

21. On page 8, lines 16-17, authors state "We also noted nonsignificant levels of Programmed Death-1+(PD1) Th17 cells in GF and ABSL-1 mice, compared to ABSL-2 mice (Figure 4c)" however, when looking at Fig4C, it is clear that ABSL-1 mice have increased levels of PD1+ cells compared to GF. Could the authors clarify what was meant in their statement?

22. On page 8, lines 19-21, IL-6+ and IL-23R+ cells are compared. In order to compare the respective diagrams, it would be easier to use the same scale for the y-axes of figures 4A and S4B.

Figure 4.

23. Panel B and caption. N number is given between 2 and 19 but there are more than 19 mice in the ABSL-2 Bleo group in panel B. Please clarify how many mice were used.

24. Panel E. From what group of mice (GF/ABSL-1/ABSL-2) are these representative histograms? It would also be interesting to have the authors interpretation of the effect of saline on IL-17A/PD-1 expression level as there are multiple peaks and a clear shift between FMO and saline, and saline and bleo

25. Panel F caption. It is stated that pSTAT3Y705 expression is measured. There is no mention of this phosphorylated amino acid in the text. Could the authors clarify in the methods why this was chosen as a marker?

26. Table 1. It is unclear why the Spearman correlation study was not performed (or is not shown) for CD4+IL-6+ T cells. If there was no correlation found, please state it in the text.

27. Page 9 from line 8. It is very interesting that GF transplanted with ABSL-2 stool show an increased weight loss and RLL collagen content. However, on Fig1b, ABSL-1 mice inoculated with bleo also lost weight, which is not found in GF mice transplanted with the ABSL-1 stool. Do the authors have any interpretation for this discrepancy?

28. In the results section the present and past tenses are used. Please homogenize the tense throughout this section to help the reader.

29. Please add ellipses for saline and bleomycin on the MDS plots as it is difficult to differentiate between the 2 groups.

Discussion

30. Butyrate is mentioned a few times in this manuscript, suggesting that it could be involved in the mechanism of action. Have the authors measured the concentration of bacterial metabolites (ie SCFA) in feces from the ABSL-1 and ABSL-2 mice?

31. Please include the paper by Yang et al (Immunity 2019 PMID: 30824326) in the discussion. This is the first paper to show that microbiota is required for bleomycin-induced lung injury. The involvement of the different IL-17 should also be discussed. Have the authors determined whether IL-17B or E positive cells are also increased in ABSL-2 vs GF or ABSL-1 mice?

Methods

32. Some techniques and protocols are missing. Please include, in the methods section, a description of the Sircol assay and Aschcroft scoring.

33. For the murine model section, please refer to the ARRIVE guidelines to include all required information

Reviewer #3 (Remarks to the Author):

The present manuscript entitled 'Gut microbiota modulates lung fibrosis severity following acute lung injury' tried to define how murine gut microbiota influence bleomycin-mediated lung fibrosis and inflammation. The authors compared bleomycin-mediated lung fibrosis and inflammation in mice from three different rearing environments and they also conducted transplantation of feces with low or high microbial diversity to prove that low microbial diversity leads to more severe lung disease.

Major points:

1. The aim of this study was to investigate the role of gut microbiota in lung fibrosis, however, the lung microbiota, which have been more extensively explored as a critical player in pulmonary fibrosis, also differ between germ-free, animal biosafety level 1, and animal biosafety level 2 mice, and may also contribute to the differences in the outcome of bleomycin treatment. Although the authors further conducted fecal transplantation, were lung microbiota influenced by fecal microbiota transplantation?
2. In Fig1a, the duration in which survival was observed was too short and no difference was found between ABSL-1 Bleo and ABSL-2 Bleo. Interestingly, no mice died in the GF Bleo group within 15 days, but how about in later time points? Based on our experiments, GF NSG mice (immunodeficient, without T, B, NK cells) develop severe fibrosis and mortality rate can be high following bleomycin instillation.
3. In Fig.2a, the authors found that species richness and evenness of gut microbiota were significantly higher in ABSL-1 mice compared with ABSL-2 mice. This seems to be a paradox to the definition of ABSL-1 (no experiments involving infectious agents; mice possess only commensal organisms), and ABSL-2 (experiments involving infectious agents of moderate potential hazard to personnel are present in environment), which indicates that ABSL-2 environment involves more microbiota species than ABSL-1 environment. Otherwise, ABSL-2 environment contains pathogens that outcompete commensal bacteria, thus reducing microbiota diversity.
4. In Fig4, the intergroup variations in the number of samples were too dramatic. Fig4b, were the GF(saline) group and GF(Bleo) group interchanged by mistake? The annotations of Fig4e are confusing.
5. In Fig5, only data on weight loss and collagen content were presented. How about survival, lung

histology and expression of inflammatory cytokines IL-6 and IL17 between these three groups ?
6. In Supplemental Fig5, these data seem to be unconvincing. Why the gating strategies were different between groups for a same marker ?

Response to Reviewers

Reviewer #1:

In this study, the authors investigated the bleomycin-induced murine model in different environment and explore the role of gut microbiota in modulating lung fibrosis via gut-lung axis, which indicated the impact of variation in gut microbial diversity on lung fibrosis severity. However, some major concerns and limitations are still needed to be elucidated.

Major comments

1. Why do you compare the environment of animal biosafety level 1 and level 2?

We apologize for not making the rationale for our study clear in the manuscript. The goal is to investigate the impact of the gut microbiome on pulmonary fibrosis. We chose three environments with distinctions in the gut microbiome: (1) uninfected C57BL/6J WT mice housed in animal biosafety level 1 (ABSL 1) non-infectious housing, and (2) uninfected C57BL/6J wildtype (WT) mice housed in animal biosafety level 2 (ABSL 2) infectious facility, and (3) uninfected germ-free mice (C57BL/6J background) in a gnotobiotic facility. Metagenomic analysis of fecal samples from the three environments [germ-free, ABSL-1 and ABSL-2] by linear discriminant analysis revealed distinct species in the ABSL 1 (*Acetatifactor muris*, *bifidobacterium pseudolongum*) and the ABSL 2 cohort (i.e., *Bacteroides thetaiotaomicron*, *Lachnospiraceae bacterium A2*) (Figure 2). As expected, the germ-free samples did not reveal distinct species. Remarkably, we noted significantly less fibrosis in the germ-free gnotobiotic cohort, and intermediate fibrosis in WT mice in the ABSL1 cohort, compared to advanced fibrosis in the ABSL 2 cohort. These observations support investigation of the hypothesis that the environment impacts the gut microbiota which in turn determines the severity of lung fibrosis pathogenesis. We have included this rationale to enhance clarity (pg. 4, lines 63-76).

2. As far as I know, it is still not clear how gut microbiota affect the lung fibrosis, but some studies have elucidated the mechanisms of how lung microbiota influence the pulmonary fibrosis (PMID: 30824326, PMID:30789747). Can you describe the relation between the gut microbiota and lung fibrosis and the difference of gut microbiota and lung microbiota in lung fibrosis?

Thank you for this insightful question, and we agree with the findings of the stellar publications mentioned above, which are now cited. We also analyzed the lung microbiome of ABSL-1 and ABSL-2 mice following bleomycin treatment and did not find any distinctions in the microbial communities (Figure 2E) Unlike the gut, the composition of the lung microbiome of mice does not differ significantly between floors (ABSL1 vs. ABSL2) or based on treatment (saline vs. bleomycin) when examined using either Bray-Curtis dissimilarities (Floor: $F_{1,37}=1.041$, $R^2=0.028$, $p=0.388$; Treatment: $F_{1,37}=0.993$, $R^2=0.027$, $p=0.397$) or the Jaccard index (Floor: $F_{1,37}=1.058$, $R^2=0.029$, $p=0.343$; Treatment: $F_{1,37}=0.733$, $R^2=0.020$, $p=0.948$ (Figure 2E-F). Akin to manuscript PMID: 30824326, we found higher Th17 cells in the lungs of ABSL2 mice and significantly less Th17 cells in the lungs of ABSL-1 mice, which contained a higher prevalence of lactobacilli in their stool (Figure 2, 4). In relation to mechanism, we now provide data demonstrating the capacity of lactobacilli to significantly reduce HLF collagen production, following stimulation via IL-17A and TGF- β 1 (Figure 6).

3. The manuscript suggested that bleomycin treatment does not alter the diversity of the gut microbiota, it is different from the previous studies (PMID 33803282, 31473797). Therefore, can you elucidate the reasons.

Thanks for highlighting those manuscripts. To our knowledge, the aforementioned studies did not account for differences in environment. We conducted our bleomycin experiments in three distinct

environments: (1) uninfected C57BL/6J WT mice housed in animal biosafety level 1 (ABSL 1) non-infectious housing, and (2) uninfected C57BL/6J wildtype (WT) mice housed in animal biosafety level 2 (ABSL 2) infectious facility, and (3) uninfected germ-free mice (C57BL/6J background) in a gnotobiotic facility. We analyzed ABSL-1 and ABSL-2 mice according to treatment and did not note a difference. We repeated the analysis of treatment but separated them by floor. That analysis revealed that treatment was associated with taxonomic composition for mice on the ABSL1 floor, but only when examining Jaccard distances (Bray-Curtis: $F_{1,17}=0.727$, $R^2=0.041$, $p=0.507$; Jaccard: $F_{2,17}=9.392$, $R^2=0.356$, $p<0.001$), and was not associated with taxonomic composition on the ABSL2 floor (Bray-Curtis: $F_{1,29}=0.577$, $R^2=0.020$, $p=0.547$; Jaccard: $F_{2,29}=0.807$, $R^2=0.027$, $p=0.561$) (Figure 2C).

4. In figure 4a,4b,4c, the number of mice models in some group is not enough to elucidate the conclusion. We agree with you. The low numerical values are in germ-free mice receiving saline. Germ-free mice have normal immunity and can respond to antigen, but have primarily naïve T cells, because they are germ-free and have not seen any antigen to drive an adaptive T cell response. When administered sterile normal saline, they again do not see any antigen to drive T cell maturation. Our flow analysis for adaptive T cells has been repeated at least 3 additional times on these very expensive mice. We have since repeated the experiments in Figure 4 to significantly increase the number of mice in each cohort, as much as possible. Each mouse serves as a single datapoint. Each cohort now ranges between 6-29.

Minor comments

1. What are the other indicators for evaluating pulmonary fibrosis?

Lung collagen content assessment by Ashcroft scoring of Trichrome blue staining and Sircol assay are standard methods to quantify collagen in murine models. Additionally, methods such as hydroxyproline and col1A immunoblot analysis are often used. In addition to the Sircol analysis, we now provide immunoblot analysis collagen fibers, such as smooth muscle actin in murine lung (Supplemental Figure 1) and col1A using human specimens (Figure 6).

2. The manuscript contains some typographical and language errors. You are advised to seek the assistance of a professional manuscript editing service.

Thank you for making us aware of the language errors. The revised manuscript has been thoroughly proofed.

3. The style of legends in figure 1 are not identical, such as (a), (b), (c), (d).

Thank you bringing this to our attention. We have gone through the manuscript and made sure that all the legends in figure 1 are identical.

4. There are some other indicators of alpha diversity as well as beta diversity, are there any difference?

We have now added Pielou's evenness and species richness as additional alpha diversity metrics and included beta diversity results based on the Jaccard index. These plots can be found in Supplemental Figures S2a and S2b. ABSL-1 reared mice were found to have higher Pielou's evenness scores (Fig. S2a). Species richness does not differ between housing environment (Fig. S2b).

Reviewer #2:

The article entitled "Gut microbiota modulates lung fibrosis severity following acute lung injury" by Chioma et al. shows that rearing environment influences the response to intranasal bleomycin

induced lung fibrosis. Germ-free mice were protected from lung fibrosis as well as mice reared in an environment providing a more diverse fecal microbiota. The study shows that in the mice with the less diverse microbiota, bleomycin administration induced more CD4+IL-17A+ T cells, which could be responsible for the more severe phenotype. Finally, after transplanting the highly diverse or the less diverse microbiota into germ-free recipient mice, bleomycin-induced lung injury reproduced some of the characteristic traits observed in the mice reared in the two distinct environments (providing highly or less diverse microbiota).

2. Overall impression of the work

This is a very interesting study providing new data about how gut microbiota can influence lung response to injury and fibrosis pathogenesis that will be of interest to others in the field. However, it would greatly benefit from additional data on the lung microbiota. Indeed, it is assumed that all the differential effects induced by the intranasal administration of bleomycin are caused by the differences in gut microbiota. It is very likely that the lung microbiota would also be different between the 3 mice groups (germ-free, ABSL-1 and ABSL-2) and could be partly responsible for the changes in response to bleomycin. The authors also performed FMT from ABSL-1 and ABSL-2 feces into GF recipient mice and recapitulate (partly) the phenotype observed in the original experiment. This suggests that the differences in phenotypes are due to changes in gut microbiota although there is no evidence showing that the microbes (or some of them) did not also colonize the lungs especially when oral gavages occurred every other for 3 weeks. I understand that these are extensive additional experiments, but I believe these data would be necessary to be able to state that “the gut microbiota modulates lung fibrosis severity following acute lung injury”.

3. Specific comments, with recommendations for addressing each comment

Major

1. Differential effect of gut and lung microbiota. It is not fully demonstrated that the changes in response to bleomycin are fully and only due to the differences in gut microbiota composition. This study would greatly benefit from including the composition of the lung microbiota in the different groups (GF, ABSL-1, and ABSL-2) and after FMT. Indeed, if fecal microbes were not able to colonize the lungs after FMT, this would strongly strengthen the role of the gut microbiota in the fibrosis pathogenesis.

This is an excellent suggestion; thank you. We now include data that unlike the microbial distinctions observed in the gut microbiome (Figure 2A-D), the lung microbiome of ABSL-1 and ABSL-2 mice does not differ significantly between floors (ABSL1 vs. ABSL2) or based on treatment (saline vs. bleomycin) when examined using either Bray-Curtis dissimilarities (Floor: $F_{1,37}=1.041$, $R^2=0.028$, $p=0.388$; Treatment: $F_{1,37}=0.993$, $R^2=0.027$, $p=0.397$) or the Jaccard index (Floor: $F_{1,37}=1.058$, $R^2=0.029$, $p=0.343$; Treatment: $F_{1,37}=0.733$, $R^2=0.020$, $p=0.948$) (Figure 2E-F). The lack of distinctions in lung microbial diversity under distinct housing conditions, compared to distinctions in ABSL-1 and ABSL-2 stool suggest that the gut microbiome is a strong influencer of lung fibrosis.

2. Animal experiments: How were the number of animals per groups determined? According to the ARRIVE guidelines, the authors should “explain how the sample size was decided and provide details of any a priori sample size calculation, if done”. This is especially important as the differences in group sizes are sometimes very important (2 to at least 25). If the authors believe that adding the n numbers on the figures would make them unreadable, this information could be included in the methods section. Please also include how many, if any, independent experiments were performed, and which

measurements/parameters were looked at for each experiment. Indeed, the n numbers also vary greatly within the same group when looking at different parameters (for example, on fig 4A, the GF saline group has 2 animals whereas it has 5 or 6 mice on Fig4B, ABSL-2 bleo group has 19 mice on fig 4A and >25 on Fig 4B...)

We agree with you. The low numerical values are in germ-free mice receiving saline. The germ-free have normal immunity and can respond to antigen. However, these mice have primarily naïve T cells, because they are germ-free and have not seen antigen to drive an adaptive T cell response. When administered sterile normal saline, they again do not see any antigen to drive T cell maturation. Our flow analysis for adaptive T cells has been repeated at least 3 additional times on these very expensive mice. Although we tried to perform all parameters on the same single cell suspension, completing a flow analysis for multiple cytokines in GF mice treated with saline was difficult due to limitations in cell number. We have since repeated the experiments in Figure 4 to increase the number of mice in each cohort, as much as possible. Each mouse serves as a single datapoint. We now include a minimum of 6 mice per cohort, but as much as 29 of the WT mice. Each cohort with more than 6 mice reflects that number of murine samples needed to conduct the experiments.

3. Impact of bleomycin on gut microbiota: It is somewhat surprising that bleomycin induced weight loss in ABSL-1 and ABSL-2 mice but did not seem to have an effect of gut microbiota composition. Intranasal bleomycin has been shown to alter lung microbiota (Yang et al Immunity 2019 PMID: 30824326) and weight loss/reduced food consumption usually induce changes in microbiota composition (Sencio et al Cell Rep 2020 PMID: 32130898)

You are correct. When we initially assessed for distinctions by treatment, we analyzed ABSL-1 and ABSL-2 mice according to treatment and did not note a difference. We repeated the analysis of treatment but separated them by floor. That analysis revealed that treatment was associated with taxonomic composition for mice on the ABSL1 floor, but only when examining Jaccard distances (Bray-Curtis: $F_{1,17}=0.727$, $R^2=0.041$, $p=0.507$; Jaccard: $F_{2,17}=9.392$, $R^2=0.356$, $p<0.001$), and was not associated with taxonomic composition on the ABSL2 floor (Bray-Curtis: $F_{1,29}=0.577$, $R^2=0.020$, $p=0.547$; Jaccard: $F_{2,29}=0.807$, $R^2=0.027$, $p=0.561$). We have now added that data (Figure 2C).

4. Statistical analysis: Most experiments have 2 variables (rearing environment and saline/bleomycin administration). Therefore, the statistical analysis should be done using a two-way ANOVA instead of a one-way ANOVA

You are correct; thank you bringing this to our attention. We have gone through the manuscript and made sure that two-way ANOVA with Tukey's multiple comparison's test where appropriate was performed. We noted no change in statistical significance following implementation of two-way ANOVA analysis of experiments (Figure 1, 4, 5, 6).

5. FMT study: Was there any metagenomics study performed on feces from transplanted animals to confirm that alpha and beta diversity indexes were similar to that of ABSL-1 and ABSL-2 mice? This is very important to have an idea about how the microbiota implanted itself in the GF recipient mice. To examine how well the microbiome transferred to germ-free mice, we used average Bray-Curtis and Jaccard distances to compare the stool samples from gavaged-mice to stool samples from mice living in their respective treatments. We compared mice gavaged with ABSL1 stool to mice gavaged with ABSL2 stool, as well as to mice from ABSL1 and ABSL2 floors. ABSL1- and ABSL2-gavaged mice did not have

statistically distinct gut microbiomes (Bray-Curtis: $F_{1,8}=2.159$, $R^2=0.236$, $p=0.091$; Jaccard: $F_{1,8}=1.426$, $R^2=0.169$, $p=0.167$), though the values approach significance and statistical power for this analysis was limited by sample size. When analyzing the stool samples of all mice (FMT and non-gavaged mice), stool origin (ABSL1 vs. ABSL2) did have a significant association with microbiome composition using Bray-Curtis dissimilarities (Stool origin: $F_{1,58}=12.378$, $R^2=0.082$, $p<0.001$) and Jaccard distances (Stool origin: $F_{1,58}=6.192$, $R^2=0.059$, $p=0.001$). Current housing (ABSL1 vs. ABSL2 vs. Germ-free) had a larger significant association with microbiome composition using Bray-Curtis dissimilarities ($F_{1,58}=45.864$, $R^2=0.607$, $p<0.001$) and Jaccard distances ($F_{1,58}=22.192$, $R^2=0.422$, $p<0.001$).

Minor

Introduction

6. Page 2, line 19, Lactobacilli and Bifidobacteria are presented as butyrate producers. Although it is true that Lactobacilli do produce butyrate, Bifidobacteria produce mainly acetate and lactate and butyrate producers are mainly Bacteroidetes and Clostridia.

Thank you for clarifying which bacterial genera produce butyrate. The text now reads “microbial metabolites, such as the short chain fatty acid (SCFA) butyrate, produced by *Bacteroidetes*, *Clostridia*, and *Lactobacilli*...” (pg. 3, lines 57-60).

Results

Figure 1:

7. Panel A. The legend is very confusing, especially between the ABSL-1 saline and bleo groups: it looks like the ABSL-1 bleo mice survived better than the ABSL-1 saline mice. Could the authors change the symbols for these groups so that the figure is read better?

Thank you bringing this to our attention. We have gone through the figure and ensured that the symbols are easily distinguishable and that the figure reads better. For clarification all the groups had 100% survival rate except for ABSL-1 and ABSL-2 bleomycin treated mice.

8. Panel B. The GF saline mice all show 0% change in weight, which is very surprising. How was the weight loss calculated? Was there any “normalization” done using the GF saline as the control group so that they would not show any weight change? It will be helpful to have this stated in the methods section.

We weigh the ABSL-1 and ABSL-2 mice at baseline and then every other day for 14 days. This is not possible with GF mice due to the possibility of contaminating them via the weighing process. To maintain their germ-free status, we weight them at baseline and again at day 14, the day of harvest. There was no normalization conducted. The observation of 100% survival (Figure 1A) and minimal fibrosis (Figure 1E) in the bleomycin treated GF mice is consistent with minimal weight loss.

9. Panel C. It is difficult to see the scale bars. Please include them on all images and make them a bit thicker to make them easier to see. Thank you for pointing this out. We have amended the figures on all images so that the scale bars are easier to see.

10. Panel E caption. It is stated that the number of mice is 6-25 but it is mentioned 2-19 in the reporting summary. Please make sure all the n numbers (for mice and independent experiments) are reported in the methods section. Thank you for pointing this out; each figure now accurately reflects the number of mice.

Figure 2:

11. Panel A. Please state in the caption what the boxes are representing (mean/median/SD...).

The boxes show the median and 25th and 75th quartiles. The whiskers extend to 1.5*IQR. We have added this verbiage to Figure 2A.

12. Panel A/C. Does one dot represent one mouse?

Yes, each dot represents one mouse. We have added this verbiage to Figure 2.

13. Panel A/B/C. Please state the number of animals used in this experiment.

Mice were used for the gut and lung microbiome analyses based upon tissue availability; there were 13-27 mice in the lung and gut microbiome analyses; this number is now included in the legend.

14. Figures 2/S2/3/S3. It is unclear why Bray-Curtis diagram is shown as beta-diversity for microbiota composition in the main figure whereas the Jaccard index diagram is shown for the beta-diversity of metagenomics functional genes. Could the authors clarify and/or homogenize which diagrams are part of the main figures and which ones are part of the supplementary files.

We apologize for the lack of clarity and have rearranged the figures accordingly. The Jaccard plots of the gut microbiome are now included in the main figures for both Figure 2 and 3, with the Bray-Curtis diagrams in the supplement. The Bray-Curtis analyses of the gut microbiome are now Supplemental Figures 3 and 4.

15. On page 6 lines 11 to 14, “ABSL-1 mice had significantly higher relative abundances of Proteobacteria and Verrucomicrobia. Increased proteobacteria abundance has been linked with both inflammatory gut and airway environments, as in Crohn’s disease, asthma and after lung transplantation”. This suggests that ABSL-1 mice would show more inflammation than ABSL-2 mice, which is not the case if we look at the results on IL-6/IL-17 further in the paper. Unfortunately, this is not discussed although it is interesting. Do the authors think that the diversity rather than the composition itself is responsible for the differences in proinflammatory responses to bleomycin?

Thank you for this comment, we agree it is an interesting observation. While an increased abundance of proteobacteria has been shown in patients with inflammatory diseases, our observations here of increased proteobacteria in our mice with less severe disease could be due to several factors. For one, the phylum proteobacteria is diverse itself with several subdivisions and includes many organisms that are human pathogens. It is possible that in our study, the less pathogenic proteobacteria is what is enriched, where in the Crohn’s disease and asthma studies, more pathogenic species are enriched. Additionally, as you suggest, diversity rather than specific taxa abundance could be driving the phenotypes we see. Evidence suggests that a less diverse gut microbiota early in life leads to an imbalanced immune system subject to the development of diseases like allergies, psoriasis, and gut inflammation (Cahenzli et al., 2013; Zanvit et al., 2015; Knoop et al., 2017). We now include these references and discuss this in the text (Pg. 14, lines 296-306).

16. Panel A. It looks like saline induced an increase in the IL-6+ T cell population in ABSL-1 mice. Do the authors have an interpretation for this? Also, is it known why the FMO profiles are different between GF/ABSL-1 and ABSL-2? We agree that there is an increase in IL-6 expression although is not statistically significant. Intranasal saline does induce mice trauma to the lung, which may heighten IL-6 and its co-receptor responses. The FMO profiles were distinct because the flow experiments were very large, and it was not possible to assess for all of the cytokines from all 3 cohorts simultaneously. We repeated an analysis from a subset and now demonstrate the same FMO profile for GF, ABSL-1, and ABSL-2 mice (Supplemental Figure 5).

17. Panel B. Looking at the overlay diagrams, saline and bleo seem to induce a bigger increase in GP130+ cells in ABSL-1 mice than ABSL-2 mice. Do the authors have an explanation for this? GP130 is the co-receptor to the IL-6 receptor. We have repeated the experiments simultaneously and the ABSL-1 and ABSL-2 is distinctly from than GF (Supplemental Figure 5).

18. Panels A/B. Selections of the positive cells, represented by the bars, seem shorter in ABSL-1 diagrams than in GF and ABSL-2 (which seem to be of similar sizes). Could the authors explain if this is a technical bias or if this has a scientific meaning?

This is no longer observed with the new experiments (Supplemental Figure 5).

19. On page 7, line 2-3: is there a particular reason for the italics on “overrepresented in ABSL-2 mice (Fig2d)”?

Thank you for catching this typographical error; the italics have been removed.

20. Figure S4. It is confusing why on panel A, there are only 2 mice in the CF saline groups and on panel B, there are only 2 mice in the GF Bleo group. Could the author clarify why the number of mice per group do not always match throughout the figures? The same is true for figure 4 panels a, b, c, and f.

We have removed panel A and included the data in Figure 4 panel A. There are now at least six mice in each cohort.

21. On page 8, lines 16-17, authors state “We also noted nonsignificant levels of Programmed Death-1+(PD1) Th17 cells in GF and ABSL-1 mice, compared to ABSL-2 mice (Figure 4c)” however, when looking at Fig4C, it is clear that ABSL-1 mice have increased levels of PD1+ cells compared to GF. Could the authors clarify what was meant in their statement?

Thank you for catching this error in our description. This statement has been replaced with “We previously reported that PD-1+ Th17 cells contribute to the development of lung fibrosis. We did note that ABSL-2 mice treated with bleomycin had significantly higher levels of PD-1+ CD4+ T cells secreting IL-17A than ABSL-1 or GF mice (Fig 4c) (Pg. 9, lines 186-192).

22. On page 8, lines 19-21, IL-6+ and IL-23R+ cells are compared. In order to compare the respective diagrams, it would be easier to use the same scale for the y-axes of figures 4A and S4B.

We have now included the data in Supplemental Figure 4A into Figure 4A with the same scale.

Figure 4.

23. Panel B and caption. N number is given between 2 and 19 but there are more than 19 mice in the ABSL-2 Bleo group in panel B. Please clarify how many mice were used.

Thank you; we note that 6-29 are in each experiment, which reflects the number of murine specimens needed to acquire the necessary data.

24. Panel E. From what group of mice (GF/ABSL-1/ABSL-2) are these representative histograms? It would also be interesting to have the authors interpretation of the effect of saline on IL-17A/PD-1 expression level as there are multiple peaks and a clear shift between FMO and saline, and saline and bleo.

The histogram was chosen from an ABSL-2 housed mouse. You are correct that there was a clear shift; however, it was not statistically significant, thus we are leery to further speculate as to any meaning it may have.

25. Panel F caption. It is stated that pSTAT3Y705 expression is measured. There is no mention of this phosphorylated amino acid in the text. Could the authors clarify in the methods why this was chosen as a marker?

pSTAT3 is a transcription factor specific for Th17 cells. We have now added text to clarify that reads as follows: "Phosphorylated STAT3 (pSTAT3Y705) indicates that the transcription factor has been activated and can induce downstream inflammatory signaling. Bleomycin treatment increases pSTAT3 expression in CD4+ T cells in the ABSL-2 cohort mice (Fig. 4f)" (Pg. 9, lines 190-197).

26. Table 1. It is unclear why the Spearman correlation study was not performed (or is not shown) for CD4+IL-6+ T cells. If there was no correlation found, please state it in the text.

We apologize for not showing the Spearman correlation. IL-6 signaling can lead to several signaling pathways, some of which have no relation with the Th17 development and subsequent. Application of the Spearman correlation revealed that there was no correlation found between CD4+ IL-6+ T cells and specific microbial taxa; this is now stated in the text and removed GP130 from the Table (Pg. 10, lines 198-206).

27. Page 9 from line 8. It is very interesting that GF transplanted with ABSL-2 stool show an increased weight loss and RLL collagen content. However, on Fig1b, ABSL-1 mice inoculated with bleo also lost weight, which is not found in GF mice transplanted with the ABSL-1 stool. Do the authors have any interpretation for this discrepancy?

This is an interesting point, and we thank you for bringing it to our attention. Microbiome analysis of FMT of ABSL-1 or ABSL-2 stool into GF mice indicates that there was not 100% transfer. We suspect that the microorganisms responsible for weight loss were not transferred during gavage of ABSL-1 stool.

28. In the results section the present and past tenses are used. Please homogenize the tense throughout this section to help the reader.

Thank you for bringing this inconsistency to our attention; the results section has been thoroughly proofed, and all verbs changed to past tense.

29. Please add ellipses for saline and bleomycin on the MDS plots as it is difficult to differentiate between the 2 groups.

Thank you for the suggestion – these have now been added to Figure 2 and 3.

Discussion

30. Butyrate is mentioned a few times in this manuscript, suggesting that it could be involved in the mechanism of action. Have the authors measured the concentration of bacterial metabolites (i.e., SCFA) in feces from the ABSL-1 and ABSL-2 mice?

Thank you for this suggestion. There are several manuscripts that demonstrate increased butyrate in lactobacilli-containing stool, and they are cited. We felt mechanism would be further demonstrated by assessing if lactobacilli supernatant has the capacity to inhibit HLF collagen production. Remarkably, we noted that it does significantly reduce HLF collagen production (Pg. 11 lines 225- 241). This data is now Figure 6.

31. Please include the paper by Yang et al (Immunity 2019 PMID: 30824326) in the discussion. This is the first paper to show that microbiota is required for bleomycin-induced lung injury. The involvement of the different IL-17 should also be discussed. Have the authors determined whether IL-17B or E positive cells are also increased in ABSL-2 vs GF or ABSL-1 mice?

We have added this reference to the discussion (Pg. 13, line 267-269). We did conduct RT-PCR analysis for IL-17B and IL-17E; it was not detected in any of the three cohorts.

Methods

32. Some techniques and protocols are missing. Please include, in the methods section, a description of the Sircol assay and Ashcroft scoring. These well-respected techniques are typically referenced. We apologize for the oversight and thank you for pointing out which techniques are missing. We have added the following text to the methods section:

“Analysis of lung fibrosis

Lung fibrosis was assessed in mice given intranasal saline or bleomycin both quantitatively and qualitatively. The right lower lobe of each lung was homogenized in acetic acid and collagen content measured using the Sircol assay (BioColor Ltd., Newton Abbey, UK) following the manufacturer’s protocol. The right middle lobe of each lung was immediately placed in formalin for fixing overnight before being transferred to ethanol prior to embedding and sectioning for histology. Slide preparation was performed by the Vanderbilt Translational Pathology Shared Resource (TPSR). Slides were stained with hematoxylin and eosin (H&E), as well as trichrome blue by TPSR. A pathologist blinded to the samples scored each slide following the updated Ashcroft scoring guidelines described by Hübner et al.⁴⁵” (Pg. 17, lines 365-373).

33. For the murine model section, please refer to the ARRIVE guidelines to include all required information Thank you for pointing us to the ARRIVE guidelines. We have incorporated them into the methods section on murine model of lung fibrosis and the legend of each figure to add clarity (Pg. 16, line 335-364).

Reviewer #3:

The present manuscript entitled 'Gut microbiota modulates lung fibrosis severity following acute lung injury' tried to define how murine gut microbiota influence bleomycin-mediated lung fibrosis and inflammation. The authors compared bleomycin-mediated lung fibrosis and inflammation in mice from three different rearing environments and they also conducted transplantation of feces with low or high microbial diversity to prove that low microbial diversity leads to more severe lung disease.

Major points:

1. The aim of this study was to investigate the role of gut microbiota in lung fibrosis, however, the lung microbiota, which have been more extensively explored as a critical player in pulmonary fibrosis, also differ between germ-free, animal biosafety level 1, and animal biosafety level 2 mice, and may also contribute to the differences in the outcome of bleomycin treatment. Although the authors further conducted fecal transplantation, were lung microbiota influenced by fecal microbiota transplantation?

That is a valid point; thank you. We now include data demonstrating no distinctions in the lung microbiome between ABSL-1 and ABSL-2 housed mice. Relative abundances of phyla in the lung microbiome did not differ significantly between mice housed on ABSL1 and ABSL2 floors (GLMM; all $p > 0.05$) (Figure 2E, F). Because of the lack of diversity in the ABSL-1 and ABSL-2 mice, we did not pursue lung microbiome analysis following FMT of ABSL-1 or ABSL-2 stool in germ free mice.

2. In Fig1a, the duration in which survival was observed was too short and no difference was found between ABSL-1 Bleo and ABSL-2 Bleo. Interestingly, no mice died in the GF Bleo group within 15 days, but how about in later time points? Based on our experiments, GF NSG mice (immunodeficient, without T, B, NK cells) develop severe fibrosis and mortality rate can be high following bleomycin instillation.

Thank you for sharing your observation. We and others note that the highest mortality typically occurs between days 6-10 following intranasal bleomycin administration. Mice that survive past day 14-21 will completely recover and heal their lung fibrosis. The accepted standard in the field is the follow mortality out to Day 14.

3. In Fig.2a, the authors found that species richness and evenness of gut microbiota were significantly higher in ABSL-1 mice compared with ABSL-2 mice. This seems to be a paradox to the definition of ABSL-1 (no experiments involving infectious agents; mice possess only commensal organisms), and ABSL-2 (experiments involving infectious agents of moderate potential hazard to personnel are present in environment), which indicates that ABSL-2 environment involves more microbiota species than ABSL-1 environment. Otherwise, ABSL-2 environment contains pathogens that outcompete commensal bacteria, thus reducing microbiota diversity.

We agree with your assessment that the organisms present in the ABSL-2 facility may outcompete the normal commensal gut flora. It is also possible that proinflammatory gut flora inhibit gut diversity as well. This observation of lower gut microbial diversity in ABSL-2 housing conditions does suggest that environment in which the host lives also impact lung disease, which is an accepted observation.

4. In Fig4, the intergroup variations in the number of samples were too dramatic. Fig4b, were the GF (saline) group and GF(Bleo) group interchanged by mistake? The annotations of Fig4e are confusing.

No, the samples are as listed. It is difficult to look for adaptive immune cells in GF mice because these mice have not seen antigen, so the immune repertoire is primarily naïve cells. We have since repeated the experiment numerous times to get a larger number of GF saline and bleomycin-treated mice and note no differences from our initial observation. We have amended Figure 4 to reflect the increase in numbers.

5. In Fig5, only data on weight loss and collagen content were presented. How about survival, lung histology and expression of inflammatory cytokines IL-6 and IL17 between these three groups?

You are correct. We now include survival and weight loss, representative histology, Sircol and flow cytometry. Unfortunately, the numbers of inflammatory cells were too low to further subgate to the actual proinflammatory cytokines, IL-17A.

6. In Supplemental Fig5, these data seem to be unconvincing. Why the gating strategies were different between groups for a same marker?

You are correct. The FMO profiles were distinct because the flow experiments were very large, and it was not possible to assess for all the cytokines from all 3 cohorts simultaneously. We repeated an analysis from a subset and now demonstrate the same FMO profile for GF, ABSL-1, and ABSL-2 mice (Supplemental Figure 6). The resultant FMO profiles are convincing.

Reviewers' comments:

Reviewer #1 (Remarks to the Author):

All of my comments were addressed adequately.

Reviewer #2 (Remarks to the Author):

The authors have performed important complementary experiments and answered most of my comments. In their new findings, it is particularly interesting that conversely to the gut, the lung microbiome was not affected by the rearing environment or the treatment with bleomycin.

In my opinion, the following points should still be addressed:

1. Although their new finding supports their conclusions from the previous version of the manuscript, I find that this result (no change in lung microbiota) should be discussed in more detail as it goes against some previously published results (by other groups) and suggests that the lung microbiota is impacted a lot less by environmental changes than the gut microbiota.
2. There are still many errors (typographical / grammar and other) in the manuscript (non-exhaustive list below). Please make sure the manuscript is carefully checked.
 - a. P4-l.79: "ABSL-2 mice administered bleomycin"
 - b. P5-l.104-105: "We confirmed these observations by performing immunoblot analysis for alpha smooth muscle actin (α -SMA) analysis..."
 - c. P6-l.107: were the animals gavaged in this part? (no mention before)
 - d. P8-l.160-161: is there a sub-heading missing? Paragraph before is about lung microbiome diversity but from l.161 it appears to be about the gut?
 - e. P9-l.194: why is "there" in italics?
 - f. P9-l.194: "There was a significant difference between ABSL-2 bleomycin and saline treated cohorts in the expression of IL-23R (Fig. S6a)" no mention of IL-23R before or after this. No conclusion regarding this result.
 - g. P10-l.217: "14 days after bleomycin injury with Scale bar"
 - h. P11-l.229 "One of the ABSL-1 species identified in greater abundance was Lactobacilli" Lactobacilli are genera not species
 - i. P11-l.231: "phase" is missing after "Lactobacillus rhamnosus in logarithmic growth"
 - j. P15-l.319-320: "however, companion microorganisms, such as lactobacilli, limit their capacity to induce lung disease severity". There is no proof for this statement in this manuscript.
 - k. P16-l.353: "The researchers performing outcome measures were blinded to the environment and treatment of each mouse". This does not fit with the reporting summary in which it is stated "Blinding was not done in this study"

Reviewer #4 (Remarks to the Author):

This paper 'Gut microbiota modulates lung fibrosis severity following acute lung injury' tried to define how murine gut microbiota influence bleomycin-mediated lung fibrosis and inflammation. The authors compared bleomycin-mediated lung fibrosis and inflammation in mice from three different rearing environments and they also conducted transplantation of feces with low or high microbial diversity to prove that low microbial diversity leads to more severe lung disease. This work is novel, largely convincing.

Editorial Comments

Your revised manuscript entitled "Gut microbiota modulates lung fibrosis severity following acute lung injury" has now been seen again by 3 referees. You will see from their comments below that while they find your work of considerable improved, some important points remain. We are interested in the possibility of publishing your study in Communications Biology, but would like to consider your response to these concerns in the form of a revised manuscript before we make a final decision on publication.

We therefore invite you to revise and resubmit your manuscript, taking into account the points raised. In addition to the comments below, reviewer 2 has communicated to the editor that it is still not reported in the Methods how the the number of animals per group were determined and the reviewer strongly suggests following the Arriva guidelines in reporting the animal experiments. We also note that although the Reporting Summary indicates all results were performed at least twice, this is not always clear from the figure legends. Please ensure to indicate how many times experiments were independently repeated in each figure legend.

Thank you; in the Methods section, we now outline how the number of mice were determined and also include data regarding the 10 major points of the ARRIVE guidelines (lines 351-369). We could not find ARRIVA; we assumed ARRIVE was what was intended. We are happy to amend it further if ARRIVA guidelines was indeed what was intended. We have also amended the figures to include the number of times the experiments were independently repeated. We include the average number of independent repeats in the figure overall as some panels requires more repeats than others due to limitations in the number of cells of interest within a given tissue specimen.

Reviewers' comments:

Reviewer #1 (Remarks to the Author):

All of my comments were addressed adequately.

Thank you.

Reviewer #2 (Remarks to the Author):

The authors have performed important complementary experiments and answered most of my comments. In their new findings, it is particularly interesting that conversely to the gut, the lung microbiome was not affected by the rearing environment or the treatment with bleomycin.

In my opinion, the following points should still be addressed:

1. Although their new finding supports their conclusions from the previous version of the manuscript, I find that this result (no change in lung microbiota) should be discussed in more detail as it goes against some previously published results (by other groups) and suggests that the lung microbiota is impacted a lot less by environmental changes than the gut microbiota.

To date, the studies demonstrating distinctions in the lung microbiota of patients with lung fibrosis have been conducted in humans. This is the first study to conduct a dual lung and gut microbiome analysis in mice. In contrast to mice, humans with lung fibrosis are not confined to a single environment, but rather are mobile which facilitates exposure to multiple distinct environments that can affect the lung microbiome. In addition, humans are exposed to other factors that affects the lung microbiome, such as tobacco smoke and antibiotics. The mice were housed in a single environment without exposure to these other mitigating factors. Concomitant lung and gut microbiome studies in humans would be revealing (Lines 263-268).

2. There are still many errors (typographical / grammar and other) in the manuscript (non-exhaustive

list below). Please make sure the manuscript is carefully checked.

a. P4-I.79: “ABSL-2 mice administered bleomycin”

We have amended the sentence to read as follows: ABSL-2 mice that were administered bleomycin exhibited the greatest mortality, and ABSL-1 survival was intermediate of GF and ABSL-2 mice.

b. P5-I.104-105: “We confirmed these observations by performing immunoblot analysis for alpha smooth muscle actin (α -SMA) analysis...”

We amended the sentence to read as follows: We confirmed these observations of higher collagen quantities in ABSL-1 and ABSL-2 mice by performing immunoblot analysis for alpha smooth muscle actin (α -SMA) analysis on lung samples acquired 14 days after bleomycin administration.

c. P6-I.107: were the animals gavaged in this part? (no mention before)

You are correct. Thank you for identifying our error. We removed gavage from this sentence.

d. P8-I.160-161: is there a sub-heading missing? Paragraph before is about lung microbiome diversity but from I.161 it appears to be about the gut?

The information is correct as listed. Gut microbiome analysis by Bray-Curtis and Jaccard is outlined on P7, lines 131-140. The analysis on lines 160-161 reflects the lung microbiome.

e. P9-I.195: why is “there” in italics?

It should have not been italicized; it has been corrected. Thank you.

f. P9-I.194: “There was a significant difference between ABSL-2 bleomycin and saline treated cohorts in the expression of IL-23R (Fig. S6a)” no mention of IL-23R before or after this. No conclusion regarding this result.

The purpose of this data is to confirm to the scientific audience who are immunologists that all components of the Th17 cell development and the IL-17A signaling pathway were assessed and are present. It is common practice to include assessments for IL-23R expression because of the role IL-23 has on Th17 cell maturation but it is so common that there is no need for further discourse.

g. P10-I.217: “14 days after bleomycin injury with Scale bar”

We have amended the sentence to read as follows: Representative images for Masson’s trichrome-stained lung histologic sections (with Scale bar) of GF, GF+ABSL-1 and GF+ABSL-2 mice are shown 14 days after bleomycin intranasal inoculation. (Fig. 5c).

h. P11-I.231 “One of the ABSL-1 species identified in greater abundance was Lactobacilli” Lactobacilli are genera not species

We agree; the genus name is *Lactobacillus*. We have replaced the word “species” with the word “microorganisms” and wrote lactobacilli in lower caps. The sentence now reads: One of the ABSL-1 microorganisms identified in greater abundance was lactobacilli, which has been reported to reduce IL-17A expression.

i. P11-I.233: “phase” is missing after “Lactobacillus rhamnosus in logarithmic growth”

Thank you; we have added the word “phase” to the sentence. The sentence now reads as follows: We examined the impact of supernatant from *Lactobacillus rhamnosus* in logarithmic phase of growth on Human Lung Fibroblast (HLF) collagen production, stimulated by the profibrotic cytokines, IL-17A and TGF β 1.

j. P15-I.326-327: “however, companion microorganisms, such as lactobacilli, limit their capacity to induce lung disease severity”. There is no proof for this statement in this manuscript.

That sentence is actually reflecting statements derived from the referenced papers, not the work that was performed by the Drake lab. We removed the word “such as lactobacilli” to minimize confusion.

k. P16-I.353: “The researchers performing outcome measures were blinded to the environment and treatment of each mouse”. This does not fit with the reporting summary in which it is stated “Blinding was not done in this study”

Thank you. The researchers performing outcome measures were blinded as much as possible throughout the entire study. We have amended the reporting summary to reflect that blinding did occur.

Reviewer #4 (Remarks to the Author):

This paper 'Gut microbiota modulates lung fibrosis severity following acute lung injury' tried to define how murine gut microbiota influence bleomycin-mediated lung fibrosis and inflammation. The authors compared bleomycin-mediated lung fibrosis and inflammation in mice from three different rearing environments and they also conducted transplantation of feces with low or high microbial diversity to prove that low microbial diversity leads to more severe lung disease. This work is novel, largely convincing.

Thank you.

REVIEWERS' COMMENTS:

1. Figure 1: These GF mouse experiments are really very difficult to do multiple times - the sample size $n = 6-25$, which is more than adequate to do stats on. The stats seem appropriate.
2. Figure 2: $n = 13-27$ mice and again for diversity analysis adequate samples. These experiments are also never repeated as the initial sample sizes are large enough to make appropriate conclusions.
3. Figure 3: same as Figure 2
4. These are flow cytometry analysis of T cells (single cells) isolated from GF mice ($n = 6-29$ mice). This is also an adequate sample size. Performing this again in a separate experiment is very hard to do both practically and cost. The sample size is large enough to make a conclusion on that experiment.
5. Figure 5. The sample size for microbiome reconstitution $n = 4-6$ is adequate -- now here, again, the argument is that it is technically and cost-wise very hard to repeat GF experiments -- if the sample size was <3 or so I would have a problem with it. I think they should acknowledge the fact that GF experiments were done once and discuss the future aspects of repeating this independently in another facility.
6. Figure 6 is unclear. $n = 5$ individual HLF were tested for coltype1a production when exposed to the various interventions. So this is one biological experiment. Here repeats would have been important using additional samples and they do not specify technical repeats. This is problematic but fixable.
7. Supplemental Figure 1. This is fine for sample size but panel "e" ANOVA should not be performed for 3 or less samples.
8. Suppl. Figure 2 is fine
9. Suppl. Figure 3&4 is fine
10. Suppl Fig 5 is unclear as to how many samples were used to generate the data. However, the fact that this came from one animal experimental set is fine given the constraints as above.
11. Suppl Fig 6 is fine for sample size

Editorial Comments

Your manuscript entitled "Gut microbiota modulates lung fibrosis severity following acute lung injury" has now been seen again by our referees, whose comments appear below. In this case, the original reviewer 2 was unavailable so I asked another microbiota expert to comment on the reproducibility aspects.

In light of their advice I am delighted to say that we are happy, in principle, to publish a suitably revised version in Communications Biology under the open access CC BY license (Creative Commons Attribution v4.0 International License).

We therefore invite you to revise your paper one last time to address the remaining concerns of our reviewers:

Please revise the statistical tests for Supplementary figure 1.

We thank the reviewer for this comment and recognize the use of $n=3$ per group to have a non-negligible risk of low reproducibility. We repeated the analysis using Bonferroni's analysis. The statistical findings were unchanged.

In addition, we are unclear what this means in the legend of figure 6: "conducted in triplicate and the average of two technical repeats was used" Please indicate the number of independent experiments.

There were five cell lines. Each individual cell line was analyzed in triplicate under each condition, and the entire experiment was conducted twice.

Also, was the experiment in figure 5 conducted twice with similar results or once? Please clarify this in the figure legend. In all cases where the experiment was performed once, please acknowledge this in the main text.

The experiment was repeated twice.

At the same time we ask that you edit your manuscript to comply with our format requirements and to maximize the accessibility and therefore the impact of your work.

Thank you; we have edited the manuscript to comply with the formatting requirements of the journal

In addition, we have been contacted by one of the co-authors, Laura Hesse, who wishes to be removed from the author list, and we would ask you to accommodate this request.

Thank you, we have removed her name from the author list

Please note that it may still be possible for your paper to be published before the end of 2022, but in order to do this we will need you to address these points as quickly as possible so that we can move forward with your paper.

REVIEWER'S COMMENTS:

1. Figure 1: These GF mouse experiments are really very difficult to do multiple times - the sample size n = 6-25, which is more than adequate to do stats on. The stats seem appropriate.

Thank you

2. Figure 2: n = 13-27 mice and again for diversity analysis adequate samples. These experiments are also never repeated as the initial sample sizes are large enough to make appropriate conclusions.

Thank you

3. Figure 3: same as Figure 2

Thank you

4. These are flow cytometry analysis of T cells (single cells) isolated from GF mice (n = 6-29 mice). This is also an adequate sample size. Performing this again in a separate experiment is very hard to do both practically and cost. The sample size is large enough to make a conclusion on that experiment.

Thank you.

5. Figure 5. The sample size for microbiome reconstitution n - 4-6 is adequate -- now here, again, the argument is that it is technically and cost-wise very hard to repeat GF experiments -- if the sample size was <3 or so I would have a problem with it. I think they should acknowledge the fact that GF experiments were done once and discuss the future aspects of repeating this independently in another facility.

Thank you; the experiments were conducted twice to achieve a total five samples. As the reviewer acknowledges, these experiments are technically difficult and expensive.

6. Figure 6 is unclear. n = 5 individual HLF were tested for coltype1a production when exposed to the various interventions. So this is one biological experiment. Here repeats would have been important using additional samples and they do not specify technical repeats. This is problematic but fixable.

Experiment was carried out using five different HLF cell lines. Each individual cell line was analyzed in triplicate under each condition, and the entire experiment was conducted twice.

7. Supplemental Figure 1. This is fine for sample size but panel "e" ANOVA should not be performed for 3 or less samples.

Thank you for pointing this out. We reanalyzed the data by Bonferroni and detected no difference in the statistical significance. The legend's statistical analysis now reflects Bonferroni.

8. Suppl. Figure 2 is fine

Thank you

9. Suppl. Figure 3&4 is fine

Thank you

10. Suppl Fig 5 is unclear as to how many samples were used to generate the data. However, the fact that this came from one animal experimental set is fine given the constraints as above.

Thank you. This data is representative of the flow cytometry data in Figure 4. One representative sample was taken from one GF, one ABSL-1 and one ABSL-2 bleomycin-treated mice. In total, this supplemental figure reflects the three mice with the most representative histograms of their particular cohort.

11. Suppl Fig 6 is fine for sample size

Thank you